# Study on the Effects of Pasture Fiber on Thermal Properties of Slag Bricks

**DOI:** 10.3390/ma17153704

**Published:** 2024-07-26

**Authors:** Zhixin Wu, Long He, Jiarui Hou, Guo Li, Jiale Ma

**Affiliations:** 1School of Architecture, Inner Mongolia University of Technology, Hohhot 010051, China; 20221800651@imut.edu.cn (Z.W.); 20231100462@imut.edu.cn (J.H.); 20211100430@imut.edu.cn (G.L.); 20221100460@imut.edu.cn (J.M.); 2Green Building Autonomous Region Key Laboratory of Higher Education, Hohhot 010051, China

**Keywords:** thermal properties, pasture fiber, recyclable composite, slag bricks

## Abstract

In the context of ecological sustainability, this study focuses on the effect of variables of pasture fibers on the thermal properties of slag bricks made from a green recyclable material. This experiment uses slag as the binder, sand as the aggregate, and pasture fiber as an additive. The experimental variables include the additive content ratio of the pasture fiber, the size of the pasture fiber, and the type of pasture fiber. Significance analysis of the experimental results of the thermal performance tests is carried out using Minitab 18.1.0 software, and the optimal ratios for the thermal performance of the composite samples are derived from the response optimizer and conformity analysis. The results of the experiment’s test analysis using Minitab 18 software indicate that, with an increase in pasture fiber content, the thermal performance of the composite samples initially decreases before increasing. Additionally, the lower the thermal conductivity of the composite sample, the lower the apparent density and the higher the porosity. Incorporating pasture fibers into slag bricks, as revealed in this study, reduces the waste of pasture resources in pastoral areas and promotes the development of sustainable building materials with favorable thermal properties.

## 1. Introduction

The continuous growth of the construction sector leads to a significant depletion of natural resources, carbon dioxide emissions, waste production, and profound ecological damage. Globally, cement production emits approximately 2 billion tons of greenhouse gases annually [1,2]. As ecological issues garner increasing global attention, utilizing waste to create new environmentally friendly building materials has emerged as a prevailing trend [3]. Incorporating waste into building bricks enhances their performance and mitigates the depletion of natural resources, fostering environmental sustainability. Many industrial wastes, including steel slag, fly ash, and gangue, can serve as raw materials for innovative green building materials. Aliabdo, Ali A., Abd Elmoaty, and Abd Elmoaty M. Emam, et al. [4], found that alkali-activated bricks composed of slag, sand, lime, and water, naturally cured at 30 °C, demonstrated optimal compressive strength. Bricks containing a higher proportion of granulated blast furnace slag (≥20%) achieved a strength of up to 35 MPa. This study indicates that industrially processed slag can meet the requirements for manufacturing building materials. To sum up, industrially reprocessed slag can meet the mechanical requirements of manufacturing building materials.

For energy sustainability considerations, enhancing the thermal performance of building materials for walls can effectively reduce energy consumption [5]. Studies demonstrate that incorporating plant fibers enhances the thermal insulation of walls [6]. Plant fibers have minimal environmental impact, positive economic benefits, low density, non-hazardous properties, and excellent insulation [7,8]. As biodegradable materials, their recycling offers opportunities for the construction industry to address ecological challenges [9]. Some researchers indicate [10,11,12,13] that incorporating plant fibers like straw, corn, and hemp in building materials creates abundant pores, enhancing their thermal performance. Moumni and Boutaina Achik et al. studied the performance of composite samples by incorporating nutshells and wheat straw into clay bricks. The findings showed that adding nutshells and wheat straw decreased the material’s bulk density and thermal conductivity, leading to higher apparent porosity and water absorption. Nevertheless, the internal porous structure improved the thermal insulation performance of the samples [6]. Furthermore, other studies have investigated the addition of jute fibers [14], straw [15], wood fiber [16], and seagrass [17] to adobe bricks. The research indicates that the characteristics of these added materials vary. For instance, straw exhibits higher water absorption, while samples containing seagrass demonstrate stronger mechanical properties. A. Laborel-Préneron et al. investigated the hygrothermal properties of composite bricks after incorporating barley straw, hemp fibers, and corn cobs into adobe bricks. The study revealed that adding a significant amount of plant fibers to adobe bricks substantially decreased the thermal conductivity [18,19,20]. There is also extensive research on incorporating plant fibers into concrete. Hua-Yueh Liu et al. found that adding sorghum powder to cement resulted in samples with lower thermal conductivity when the sorghum powder reached the optimal mixing ratio [21]. Bouasker and Marwen Belayachi et al. investigated the addition of wheat straw to lightweight aggregate concrete. Through measurements of the sample’s porosity and moisture absorption behavior, they found that this composite material exhibited a very low bulk density, high water absorption capacity, and excellent moisture regulation ability [22]. Research on the addition of natural fibers to concrete also includes flax, hemp, coconut shell fibers, jute, bamboo, coconut, seaweed, sisal, abaca fibers, banana fibers, ramie fibers [23,24,25,26,27,28,29,30,31], rice husks [32], straw and microcrystalline cellulose [33], sunflower [34], lavender [35], pasture fiber, and rubber powder [36], among others. Although studies on incorporating plant fibers into earth bricks and concrete have been very diverse, there are few studies on incorporating pasture into earth bricks and concrete. The only one is an analysis of the addition of pasture fibers to rubber powder concrete to study the rate of loss of compressive toughness and softening properties of this composite material. In summary, there are very few studies on the incorporation of pasture fibers into the new green material slag to enhance its thermal properties.

Pasture fiber is rich in cellulose, hemicellulose, and lignin. Adding alfalfa to adobe bricks can enhance their thermal performance [37]. Therefore, the study conducted in this paper is based on natural pasture fibers to investigate the effect of pasture fibers on the thermal properties of slag bricks as a new green material.

Inner Mongolia is an autonomous region in northern China characterized by vast and open terrain boasting expansive grasslands. In 2018, the output value of animal husbandry in Inner Mongolia accounted for 43% of the total output value, second only to agricultural output. By 2023, the area of cultivated pasture crops such as corn, alfalfa, fodder oats, and pasture grass in Inner Mongolia exceeded 18 million mu (about 1.2 million hectares), with a yield exceeding 20 million tons. Additionally, the production of natural grassland pasture fiber exceeded 30 million tons. However, most pasture fiber naturally withers and decomposes on the grasslands, with only a tiny portion used as animal feed—the large-scale production results in significant waste. Introducing pasture fiber into the production of building materials presents a new direction for using pasture fiber resources.

This paper explores pasture fibers from Inner Mongolia as additives to alkali-activated slag blocks; evaluates the effects of pasture fiber type, content, and size on the thermal properties of slag composite blocks; and seeks the optimal ratios for thermal properties. The researchers mixed granulated blast furnace slag, national standard sand, lime, water, and pasture fibers in fixed proportions to produce lightweight bricks. The experimental measurements show that, at a 3:1 ratio of granulated blast furnace slag to sand and a temperature of 15 °C, the compressive strength of slag bricks reaches 14 MPa without any pasture fiber additives. Building upon this, different proportions (2%, 4%, and 6%) and types (alfalfa, millet-like sprangletop, and sorghum) of pasture fiber, with varied sizes (1 mm, 2 mm, and 3 mm), were added to investigate their effects on the thermal performance of the composite samples.

## 2. Materials and Methods

### 2.1. Raw Materials

This experiment’s slag originates from ground-granulated blast furnace slag (GGBFS) manufactured by Longze Water Purification Materials Co., Ltd. in Gongyi City, China. It has a 2.9 g/cm^3^ density and a strength grade of S105. Table 1 shows its chemical composition and the structural parameters, which are compliant with GB/T18046-2017 [38]. These components were obtained through the XRF technique.

Table 2 shows that GGBFS contains the highest proportion of SiO_2_ and CaO, followed by Al_2_O_3_. GGBFS is primarily composed of the CaO-SiO_2_-Al_2_O_3_-MgO system, defined as a mixture of magnesium silicate (2CaO MgO·2SiO_2_), calcium aluminosilicate (2CaO Al_2_O_3_ SiO_2_), and depolymerized calcium aluminosilicate glass. The cations Si^4+^ and Al^3+^ play a crucial role in forming the critical glass network, while Ca^2+^ and Mg^2+^ act as modifiers in the presence of alkalis. Observations using SEM micrographs revealed that the particles of GGBFS predominantly exhibit a boxy shape, with a minor presence of spherical particles. The pasture fiber used in this study was collected from pastures in the western region of Hohhot, Inner Mongolia. After passing through sieves with mesh sizes of 18, 10, and 8, it underwent sorting into different size gradients of 1 mm, 2 mm, and 3 mm. The sieves used for this experiment were customized experimental sieves from JiuFeng Sieve, and Figure 1 shows the sieve hole apertures in 8-mesh, 10-mesh, and 18-mesh shapes. To separate the pasture sizes, the collected pasture was poured into an 8-mesh sieve and shaken manually from side to side for approximately 30 s to sift the pasture. Then, the sifted pasture was transferred to a 10-mesh sieve, which was shaken horizontally by the same person for about 3 min. After three minutes, the pasture on top of the sieve was set at the experimental size of 3 mm. The same procedure was repeated, and, after three minutes, the pasture on top of the sieve was of the experimental size of 2 mm and the sieved pasture was of the experimental size of 1 mm. Figure 2, Figure 3 and Figure 4 show the pasture fiber used in this study. The river sand used in this study was sourced locally in Hohhot and meets the production standard of GB/T 17671 [39].

### 2.2. The Tests Applied to the Samples

Initially, the crushed pasture fiber straw underwent sieving using standard sieves with apertures of 18 mesh (1 mm), 10 mesh (2 mm), and 8 mesh (3 mm). The researchers categorized the pasture fiber straw into three size gradients of 1 mm, 2 mm, and 3 mm. The experiment identified alfalfa, millet-like sprangletop, and sorghum as the types of pasture fiber used. Pastoral areas commonly have three types of structures that residents widely utilize. The sand used in the experiment was sieved based on the requirements of Chinese construction standards, with a screening criterion of less than 4.75 mm. The S105 grade slag powder and sand were mixed in a ratio of three to one.

Additionally, lime was added in a ratio of 1:8 with slag to enhance the material’s adhesive properties and achieve optimal results. Adding lime has also been demonstrated to prevent issues such as the decay of the pasture fiber material during the curing period. Different types and sizes of pasture fiber are blended into a mixture of slag, sand, and lime at proportions of 2% (5 g), 4% (10 g), and 6% (15 g). After thorough mixing, 130 g of water is added to each mixture to achieve a uniform paste consistency after stirring for 3–4 min. The uniformly mixed paste-like mixture is poured into molds measuring 20 × 20 × 70 mm. To ensure the dense filling of the paste, an oil plasterer’s trowel is used to manually tamp the mixture uniformly in a spiral motion from the edges towards the center, tamping 25 times. During tamping, if the mortar settles below the top of the mold, additional mixture is added promptly, ensuring that the mixture extends to 6–8 mm above the top surface of the mold. After waiting for 2–3 h, the excess paste is scraped off the surface using a ruler. After a 48-h settling period at a room temperature of 15 ± 2 °C, the samples are removed from the molds and transferred to a controlled indoor environment with the same temperature for curing. During the 28-day curing period, the specimens are moistened daily to ensure uniform wetting of all six surfaces. Figure 5 shows the samples after curing.

### 2.3. The Complete Factorial Design

The complete factorial design is a direct and systematic approach to experimental design, evaluating the effects of single factors and the interactions among the different factors [40]. As the factor levels or the number of variables increase, the number of required test points for this design grows exponentially. A 2-k factorial design constitutes a subset of a complete factorial design, where each factor is restricted to two discrete levels. Using this methodology, the researchers examine the main effects (impact of independent variables on the dependent variable) and interactions (influence of interactions between independent variables on the dependent variable) of both categorical and continuous components [41]. This study identified three factors with three levels each (3 × 3 × 3), and their specifications are provided in Table 3.

The experimental design comprises 27 experiments based on a complete factorial design, with three factors and three levels, replicated twice. A control group of slag bricks without added pasture fiber straw exists. Table 4 presents the results of the experimental design.

An analysis was conducted using Minitab 18, a statistical software tool, for desirability analysis and analysis of variance (ANOVA) for all combinations. A confidence interval of 95% was selected to determine the significant levels of pasture fiber type, content, and size, as well as their corresponding F-values and contribution percentages. These results are organized into a table for reporting purposes.

### 2.4. Test Methods

#### 2.4.1. Characterization of Raw Materials

Scanning electron microscopy (SEM, JEOL Ltd., Tokyo, Japan) was utilized for the microstructural characterization of the pasture fiber. The powdered forms of three types of pasture fiber (alfalfa, millet-like sprangletop, and sorghum) were examined using a HITACHI S3400N (Tokyo, Japan) scanning electron microscope. The SEM has a resolution of 3.0 nm for secondary electrons and 4.0 nm for backscattered electrons, with an acceleration voltage ranging from 300 V to 30 kV. SEM images of the three different types of pasture fiber are presented in Figure 6, Figure 7 and Figure 8. The SEM images exhibit the intrinsic fibrillation structure of cellulose fibers with varying diameters in millet-like sprangletop. These fibers aggregate into clusters, forming a honeycomb-like structure with micrometer-sized particles adhering to the periphery. The sorghum fibers display a lamellar structure characterized by folded paper-like bundles, with smaller particles discretely distributed in the vicinity. Under microscopic examination, the alfalfa fibers reveal a densely packed cluster structure, where spherical particle aggregates amalgamate to form conglomerates of varying sizes, displaying intricate interactions. Figure 9 depicts the XRD diffraction patterns of the three different types of pasture fiber. Research by Garrouri, Sabrine Lakhal, Wissem Benazzouk, and colleagues demonstrates that pasture grass primarily consists of cellulose, with hemicellulose and lignin as secondary components.

#### 2.4.2. Physical Properties

The density of a material intricately links to its thermal conductivity, which predominantly determines its thermal performance. The material density, in turn, is influenced by sample volume, mixture ratios, and fiber content. Moreover, porosity and water absorption properties further modulate density, affecting the material’s internal structure and, consequently, its thermal behavior. Therefore, investigating the physical properties of composite samples is indispensable for comprehending the factors that influence their thermal performance.


*Apparent Density*


Apparent density refers to the ratio of a material’s mass to its apparent volume in a dry state, where the apparent volume includes both the solid volume and the volume of closed pores. This study utilized the method outlined in the JGJ70-2009 standard [42] for determining mortar density. The pasture-fiber–slag composite samples underwent continuous oven drying at approximately 105 ± 5 °C until a stable mass was achieved within a 2-h period, after which the mass was measured. Subsequently, the dimensions of the samples were measured using a vernier caliper, and the sample volume was calculated accordingly. The formula for calculating the apparent density is as follows:(1)$$\rho =\frac{m}{V_{0}}$$

In the equation, m represents the mass of the sample after drying and V0 denotes its apparent volume.


*Water Absorption*


Water absorption is the percentage of water absorbed by a material when saturated relative to its dry mass. The JGJ70-2009 standard method was employed to determine water absorption in this study. After removing the cured samples, they underwent a drying process at 105 ± 5 °C for 48 ± 0.5 h and were then weighed. Subsequently, the samples were submerged in a water tank with a water level at least 20 mm above the upper surface, maintaining the water temperature at 20 ± 2 °C. Following a soaking period of 48 ± 0.5 h, the samples were removed, excess surface water was removed with a damp cloth, and the samples were weighed again. The formula for calculating the water absorption is as follows:(2)$$Wa=\left(\frac{W_{2}-W_{1}}{W_{1}}\right)$$

In the equation, W_a_ represents the water absorption (%), W_1_ is the weight of the sample after drying (g), and W_2_ is the weight of the sample after wetting (g).


*Porosity*


Porosity refers to the percentage of void volume in a lumped material compared to its total volume in its natural state. Table 5 displays the experimental measurements of the true density and apparent density. The true density of the composite samples was determined using a true-density electronic densimeter. The formula for calculating the porosity is as follows:(3)$$P_{a}=\frac{\rho_{2}-\rho_{1}}{\rho_{2}}$$

In the equation, P_a_ represents the porosity (%), ρ_2_ is the true density of the composite sample (g/cm^3^), and ρ_1_ is the apparent density (g/cm^3^).

#### 2.4.3. Thermal Properties

The thermal performance changes in pasture-fiber–slag composite samples were studied through measurements and calculations of thermal conductivity, thermal diffusivity, and specific heat capacity. The thermal conductivity, thermal diffusivity, and specific heat capacity of pasture fibers were determined using Hot Disk TPS3500 equipment. The experimental apparatus is depicted in Figure 10 [43]. The Hot Disk equipment employs transient plane source (TPS) technology to determine the thermal performance parameters. This method involves injecting the probe with constant electrical power while monitoring the temperature change in the sample [44]. The experiments were conducted at an ambient temperature using the Hot Disk apparatus, with the probe being positioned between two identically sized cubic samples measuring 70 × 70 × 20 mm. The electrical resistance of the two sensors was recorded as 6.97 Ω and 6.89 Ω, respectively. The electrical power ranged from 10 mW to 20 mW, and each measurement lasted 10 s. Two samples were tested per experimental group, with three measurements taken and averaged for the final value. Before testing, the sample surfaces were polished to ensure optimal contact with the TPS probe, serving as a temperature sensor and a heat source. The composite samples’ thermal diffusivity, specific heat capacity, and thermal conductivity were derived from mathematical models based on temperature and probe response time records.

## 3. Results

### 3.1. The Apparent Density and Porosity of the Composite Samples

The apparent density of the experimental samples exhibits variations in response to changes in pasture fiber content, size, and type, as depicted in Figure 10 and Figure 11. It is evident from the graph that the type, content, and size of the pasture fiber exert distinct influences on the apparent density of the composite samples. With a constant ratio of solid to liquid, slag, and sand, an increase in the added pasture fiber content from 5 g to 15 g results in a decline in the apparent density of the composite samples from 1.57 g/cm^3^ to 1.27 g/cm^3^, representing a decrease of 19.1%. The apparent density of the composite samples decreases with an increase in pasture fiber content.

The apparent density range for the alfalfa composite samples varies from 1.19 g/cm^3^ to 1.57 g/cm^3^. In contrast, millet-like sprangletop samples range from 1.2 g/cm^3^ to 1.45 g/cm^3^, and those with sorghum range from 1.29 g/cm^3^ to 1.4 g/cm^3^. Compared to the experimental samples without any pasture fiber additives (denoted as M0S0), the addition of alfalfa, millet-like sprangletop, and sorghum resulted in maximum reductions in the apparent density of 22.7%, 22.1%, and 16.2%, respectively. Furthermore, in samples with a pasture fiber addition of 5 g, the apparent density values for alfalfa range from 1.53 g/cm^3^ to 1.57 g/cm^3^, for millet-like sprangletop from 1.43 g/cm^3^ to 1.45 g/cm^3^, and sorghum from 1.35 g/cm^3^ to 1.38 g/cm^3^. Based on these findings, adding 5 g of alfalfa may not substantially alter the apparent density of composite samples. However, the variation in alfalfa content exerts the most significant influence on composite sample density among the three types of pasture fiber. Conversely, the impact of sorghum is minimal. In contrast, the apparent density is less affected by the size of the pasture fiber.

The porosity of composite samples is closely related to their density. With a fixed volume, a lower density results in higher porosity. Increased porosity reduces material mass, decreases mechanical strength, and heightens water absorption, ultimately compromising material durability.

Figure 12 depicts the variations in the porosity of the composite samples resulting from changes in pasture fiber content and size. These values fluctuate within the range of 39% to 52.64%. The porosity of the samples without pasture fiber is 32.67%. With 5 g of pasture fiber, the porosity increases to 43.17%, representing a 33.8% rise. For 10 g, the porosity reaches 47.15%, showing a 43.4% increase. For 15 g, the porosity reaches 48.7%, indicating a 49.1% increase. The addition of pasture fiber leads to the generation of numerous pores in the slag bricks. With a higher pasture fiber content, the porosity of the composite samples increases accordingly. This rise in porosity contributes to the enhanced thermal performance of the composite samples.

Furthermore, when the pasture fiber content is 5 g, the porosity of the composite samples with added alfalfa is 39.9%, with millet-like sprangletop is 44.0%, and with sorghum is 45.6%. The range of variation falls between 39.2% and 47.5%, with a difference of 8.3%. For a pasture fiber content of 10 g, the porosity of the composite samples with added alfalfa is 46.1%, with millet-like sprangletop is 48.2%, and with sorghum straw is 47.2%. The range of variation is between 46.1% and 48.2%, with a difference of 2.1%. With a pasture fiber content of 15 g, the porosity of the composite samples with added alfalfa is 48.7%, with millet-like sprangletop is 48.4%, and with sorghum is 49.0%. The range of variation is between 48.4% and 49.0%, with a difference of 0.6%. Therefore, when the pasture fiber content is 5 g, the influence of pasture type on porosity is pronounced. However, as the pasture fiber content increases, the impact of pasture type on the porosity of the composite samples becomes inconclusive. The composite samples containing added alfalfa exhibited the lowest pore formation compared to the other two types of pasture fiber, at 44.9%; conversely, the inclusion of sorghum resulted in the highest pore formation, at 47.27%, as depicted in Figure 13.

Figure 14 presents the inverse relationship between the apparent density and porosity. As the content of pasture fiber increases, the apparent density decreases noticeably, and the porosity increases correspondingly. This can be attributed to the lower density of pasture fiber and the significant formation of pores during the sample preparation process, which reduces the weight of the composite samples.

### 3.2. Water Absorption and Porosity of the Composite Samples

Figure 15 illustrates the influence of varying levels and sizes of pasture fiber content on the water absorption of the composite samples.

With the pasture fiber content increasing from 5 g to 15 g, the water absorption rises from 28.5% to 34.6%. The water absorption increases with the rising pasture fiber content. However, this phenomenon varies notably depending on the specific type of pasture fiber added. Among the three types of pasture fiber, the changes in water absorption are more pronounced in the composite samples with added alfalfa and millet-like sprangletop. The former increased from 24.58% to 35.25%, representing a 43.4% increase, while the latter increased from 29.12% to 34.69%, indicating a 19.13% increase. However, the water absorption of the composite samples with added sorghum ranged from 31.79% to 33.79%, showing relatively minor fluctuations. Moreover, based on the graph, it is apparent that the composite samples containing added sorghum demonstrate superior water absorption capability, registering at 31.8%. Conversely, the composite samples enriched with alfalfa exhibit a lower water absorption capacity, recording 24.6%. This is illustrated in Figure 16.

The composite sample J15S1′s water absorption is 48.74%. This is because, after being immersed in water for 6 h, the brick completely disintegrated, resulting in the material becoming granular and scattered. This resulted in the addition of 15 g of millet-like sprangletop, not meeting the requirements for construction materials. Upon error correction of the water absorption data, it was determined that the value of sample J15S1 lacked reliability. Therefore, this value was excluded from the calculation process.

The relationship between the type and size of pasture fiber and the water absorption of the composite samples is illustrated in Figure 17. The water absorption of composite samples with varying sizes of alfalfa ranged from 30.52% to 31.08%, while those with different sizes of millet-like sprangletop ranged from 32.57% to 32.7%. The water absorption of composite samples with different sizes of sorghum ranged from 31.24% to 33.36%. Therefore, the size of the pasture fiber had no significant effect on the water absorption of the composite samples. The water absorption of the composite sample with 5 g of alfalfa addition was 24.6%, which was lower than that of the slag bricks without any pasture fiber addition, which was 28.3%. This suggests that pasture-fiber–slag composite samples with no more than 5 g of pasture fiber addition exhibit reduced water absorption, potentially indicating improved durability.

Figure 18 illustrates the relationship between porosity and water absorption in composite samples with varying amounts of pasture fiber. It is evident that the fluctuation pattern of water absorption and porosity aligns, indicating that higher water absorption corresponds to higher porosity in the composite samples. However, this trend is less pronounced when the added amount of pasture fiber is 5 g.

### 3.3. The Thermal Properties of Composite Samples

Thermal conductivity is a crucial and fundamental thermophysical property. The coefficient of thermal conductivity is influenced by factors such as the size, number, and shape of pores within the material and the material’s temperature, humidity, and density. The thermal conductivity of the composite samples obtained in the experiment varied widely, ranging from 0.17 W/mK to 0.38 W/mK. Table 6 presents the thermal performance values of all composite samples. The experimental results indicate that the addition of pasture fiber has a positive impact on the thermal performance of slag bricks. Figure 19 illustrates the relationship between pasture fiber content size variation and the thermal conductivity of the composite samples. It can be observed that, when the pasture fiber content is 5 g, the thermal conductivity of the composite samples ranges from 0.27 W/mK to 0.28 W/mK; when the pasture fiber content is 10 g, the thermal conductivity of the composite samples ranges from 0.19 W/mK to 0.22 W/mK; and when the pasture fiber content is 15 g, the thermal conductivity values of the composite samples range from 0.21 W/mK to 0.25 W/mK. The data indicate that, as the pasture fiber content increases, the thermal conductivity of the composite samples initially decreases before exhibiting an upward trend. The impact of pasture fiber size on the composite samples’ thermal conductivity is minimal. While there is an increase in thermal conductivity with larger fiber sizes, the extent of this change is insignificant.

As shown in Figure 20, the composite samples with added alfalfa exhibited the slightest decrease in thermal conductivity, decreasing by 39% among the three types of pasture fiber. On the other hand, composite samples with added sorghum showed the most significant decrease in thermal conductivity, dropping by 48.5%. The addition of sorghum forms more pores within the composite sample, enhancing its thermal insulating properties. As shown in Figure 21, it is clear that the thermal conductivity of composite samples incorporating three types of pasture fiber initially decreases and then increases with the rise in pasture fiber content. At an addition of 10 g of alfalfa, the composite material achieves its lowest thermal conductivity at 0.23 W/mK, which then increases to 0.24 W/mK with an addition of 15 g of alfalfa. The thermal conductivity of the composite samples reached a minimum of 0.19 W/mK and 0.20 W/mK with additions of 10 g of millet-like sprangletop and sorghum, respectively. This represents a reduction in thermal conductivity of 56.6% and 54.1%, respectively, compared to the slag bricks without any pasture fiber addition. The thermal conductivity of the composite samples with millet-like sprangletop and sorghum is 0.21 W/mK and 0.23 W/mK, respectively, with an additional amount of 15 g. This represents an increase of 12.8% and 16.2% compared to the extra 10 g. Pasture fiber increases the porosity inside of the composite samples, resulting in reduced thermal properties. However, once the content of pasture fiber reaches a certain threshold, the thermal conductivity increases instead, adversely impacting the composite samples’ thermal properties.

Table 7 presents the composite samples’ mean thermal conductivity values, apparent density, and porosity with varying contents. Based on the data shown in the figure, the thermal conductivity decreases with increasing porosity, meaning that a lower density results in lower thermal conductivity. However, when the density is below a certain threshold, a further increase in porosity increases the thermal conductivity. This is because excessive porosity results in an increased number of pores and larger pore sizes.

In building thermal engineering, the greater the thermal storage coefficient, the better the material’s thermal stability. For colder regions, houses must be adequately insulated to ensure a comfortable indoor thermal environment in winter. Rooms with different properties of use have different thermal performance requirements. Rooms that are used throughout the day require better thermal stability, while rooms that are only used for a period of time may not need to have such good thermal stability. The thermal stability of a room depends on the thermal stability of the internal and external envelope. For rooms with high thermal stability requirements, the materials on the inner side of the envelope should have better thermal storage and larger thermal inertness index values, i.e., materials with a higher density and a more significant thermal storage coefficient are preferred for construction. For rooms with general thermal stability requirements, the materials inside of the envelope should be constructed with a preference for materials with lower density and lower thermal storage coefficients. The variation of the thermal storage coefficient of the composite samples is shown in Figure 22. The heat storage coefficient is calculated according to the following equation:(4)$$S_{24}=0.27\sqrt{\lambda c\rho }$$

In the equation, S_24_ represents the thermal storage coefficient of the material over a 24-h temperature fluctuation period, λ denotes the material’s thermal conductivity, c stands for the material’s specific heat capacity, and ρ represents the density of the material.

The composite sample with alfalfa addition of 5 g had the highest thermal storage coefficient, followed by millet-like sprangletop. In other words, among all of the composite samples with pasture fibers added, the composite sample with 5 g of alfalfa and millet-like sprangletop added had the best thermal stability. Comparing the thermal storage coefficients of other building materials, the thermal storage coefficient of all-lightweight concrete with a 1.3 g/cm^3^ density is close to 5.98 W/m^2^K. From the thermal storage coefficient value, the composite sample with 5 g of pasture fiber has the potential to act as an excellent thermal insulation material.

### 3.4. Factor Analysis and Optimization of Composite Sample Properties

The objective of employing factor analysis was to discern the significant effects of three factors—pasture fiber type, content, and size—on the apparent density, water absorption, porosity, and thermal properties of the pasture fiber slag composite samples and to determine the optimal combination through prediction using a response optimizer. A complete factorial design was performed on all composite samples with various mixing levels, as shown in Table 3. The responses, such as the apparent density, water absorption, porosity, thermal conductivity, thermal diffusion coefficient, and specific heat capacity, were subjected to ANOVA using Minitab 8.1.0 software. A confidence interval of 95% was applied. The significance of each factor was determined based on the *p*-value (considered significant if *p* < 0.05, otherwise deemed not substantial).

Additionally, the percentage contribution of each factor and interaction was calculated and is presented in tabular form. The percentage contribution reflects the proportion of total variation (SS) attributed to each factor and interaction. Table 8 presents the degrees of freedom, *p*-value, significance, and percentage contribution of the three factors and two-by-two factor interactions to the response analysis. The table reveals that the pasture fiber type and content significantly affect the composite samples’ thermal properties.

In contrast, the size of the pasture fiber shows no significant effect on the composite samples’ thermal conductivity or specific heat capacity, as indicated by a *p*-value > 0.05. These statistical results suggest that pasture fiber type, content, and size significantly influence the composite samples’ apparent density, water absorption, and porosity. Additionally, the contribution of pasture fiber content to thermal conductivity, specific heat capacity, apparent density, porosity, and water absorption was found to be the highest among all single and interaction factors. The percentage contributions are as follows: thermal conductivity, 37.48%; specific heat capacity, 20.47%; apparent density, 65.18%; porosity, 49.07%; and water absorption, 46.58%. These data indicate that pasture fiber content significantly influences the performance of the composite samples. These data suggest that pasture fiber content is critical in controlling the performance of composite samples.

These statistical results further confirm that pasture fiber content and species significantly impact the thermal properties of the composite samples. Conversely, the size has minimal influence on these properties, as indicated by the insignificant effect of size variations on thermal conductivity and specific heat capacity (*p*-value > 0.05). In the table, the ratio of the contribution of the interaction between the factors to the development of the properties of the composite samples indicates that the interaction of pasture fiber type and size does not significantly affect the thermal conductivity. Similarly, the two-way interactions of pasture fiber type and content, as well as content and size, do not significantly influence the specific heat capacity of the composite samples.

The primary and interaction effect plots of apparent density, water absorption, and porosity are depicted in Figure 23, Figure 24, Figure 25, Figure 26, Figure 27 and Figure 28. From the interaction plot of apparent density, it is evident that the apparent density of composite samples decreases with an increase in pasture fiber content. The apparent density remains unaffected by the size of the fibers when the addition ranges between 10 g and 15 g. When adding 5 g of pasture fiber, the apparent density shows less sensitivity to the fiber size. When adding three different types of pasture fiber, the apparent density of the composite samples with 5 g of alfalfa addition was the highest, exceeding 1.50 g/cm^3^. At the same time, sorghum had the lowest, at 1.35 g/cm^3^. Conversely, when the addition amount was 15 g, alfalfa exhibited the lowest apparent density, at 1.20 g/cm^3^, while sorghum showed the highest, at 1.3 g/cm^3^. The apparent density of the composite samples exhibited the most substantial variation with increasing content for added alfalfa, while sorghum demonstrated the slightest variation.

Water absorption increased with rising pasture fiber content. Notably, sorghum exhibited minimal variation with increasing content. In contrast, both alfalfa and millet-like sprangletop displayed similar, significant changes in magnitude, with their slopes in the line graphs of water absorption being comparable. Additionally, the influence of pasture fiber size on water absorption was found to be insignificant.

The porosity increased with the rise in pasture fiber content. The change in porosity was most pronounced with varying alfalfa content, ranging from less than 40% to 49%. In contrast, sorghum showed the slightest variation, maintaining levels between 45% and 50%. Additionally, no significant relationship was observed between pasture fiber size and the composite samples’ water absorption.

Figure 29 and Figure 30 show the main and interaction effect plots for the thermal conductivity of the composite samples. From the interaction plot of the thermal conductivity of the composite samples based on content and type, it is evident that the thermal conductivity of the composite samples initially decreases and then increases with the increase in pasture fiber content. Among the composite samples, those with millet-like sprangletop addition exhibited the lowest thermal conductivity at 10 g of pasture addition, followed by sorghum, while alfalfa had the highest thermal conductivity. Furthermore, while the thermal conductivity of the composite samples with added alfalfa increased slightly when the amount was raised from 10 g to 15 g, those containing millet-like sprangletop and sorghum exhibited more pronounced increases. Particularly noticeable was the case of sorghum, where the composite samples achieved their lowest thermal conductivity at the addition of 10 g, and the thermal conductivities of the two composite samples were nearly equal at 5 g and 15 g of addition. The effect of size on the thermal conductivity of the composite samples is insignificant.

Multi-response optimization combinations and predictive analyses are conducted, and the weights and importance are determined based on responses and prior experiences. Table 9 presents the specific values. Figure 31 displays the optimal combination determined using the response optimizer and desirability analysis for the pasture fiber slag composite sample with a 5 g addition of sorghum and a pasture fiber size of 1 mm. Table 10 lists the obtained prediction and validation results.

### 3.5. Microstructural Analysis of Composite Samples

This paper analyzes and presents the microstructure of composite samples based on adding pasture fiber with different contents and types. The aim is to support the study of pasture fiber slag composite samples’ physical and thermal properties by the microanalysis of composite samples. After preservation, the experimental samples were cut to the required size for testing equipment. Scanning electron microscopy was conducted on the composite samples to examine their internal behavior. It can be observed from Figure 32 that there are a large number of pores inside each sample. These pores are of different sizes, and the addition of pasture fiber makes more pores in the area where the pasture fiber is connected to the slag. In addition to this, the hydration reaction of the slag also produces a large number of pores inside of the material, which improves the thermal properties of the composite samples. Cracks can be seen in Figure 32c, which may have formed due to the incomplete reaction of some slag particles during the process, thereby compromising the structural integrity of the composite sample.

Figure 33 displays microstructural images of the composite samples with added pasture fiber slag. Figure 33(1-a–1-c) depict microscopic images of alkali-excited slag bricks without added pasture fiber, magnified at 500×, 1000×, and 3000×, respectively. Figure 33(4-a–4-c) show microstructural images of the composite samples with added pasture fiber contents of 5 g, 10 g, and 15 g, respectively, magnified to the abovementioned levels. With the incorporation of pasture fiber, the samples exhibited a discernible pore structure, which became denser with a greater pasture fiber content. This porous morphology contributed to the improved thermal properties of the composite samples. The highest porosity, reaching 52.15%, was noted by adding 15 g of pasture fiber. This observation aligns with the calculated porosity values. As observed in the images magnified 3000 times, the samples without pasture fiber addition exhibited dense structural characteristics. However, increased pasture fiber content made the samples more porous, resulting in a sparser particle-to-particle structure and poorer densification. Figure 33(5–7) depict microscopic images of samples with different types of pasture fiber added at the same amount. At 500× magnification, the images illustrate an increase in pore density, due to the addition of various pasture fibers. However, their microstructures remained distinguishable, with the composite samples containing added alfalfa exhibiting flatter, denser planes, while those with sorghum added showed more pronounced concave and convex undulations. When magnified to 3000×, the composite samples with sorghum added had bulkier clumps in their microstructure, which possessed more pores and lower thermal conductivity than the tightly packed structure of homogeneous particles.

## 4. Conclusions

This research uses pasture fiber as an additive in developing new pasture fiber slag composite bricks through compete factorial design. This study investigates the impact of altering the type, content, and size of pasture fiber on the thermal and related properties of the composite samples. We perform multi-response optimizer and statistical analysis to obtain the best combination. Micro-analysis is carried out in order to understand the potential of composite bricks as a novel material. The summary and critical conclusions based on the observations are given below, as follows:

As pasture fiber content increases, the apparent density of the slag–pasture composite samples decreases, while porosity increases, water absorption increases, and thermal conductivity first decreases and then increases.

Changes in the content of added alfalfa had the most significant effect on the apparent density of the composite samples, while sorghum had the slightest effect. The impact of pasture fiber types on porosity was substantial when the amount of pasture fiber added was 5 g but became gradually insignificant as the content of pasture fiber increased. The thermal conductivity of the composites reached its minimum value at a pasture fiber addition of 10 g. Millet-like sprangletop exhibited the lowest thermal conductivity, followed by sorghum, with alfalfa having the highest value.

The effect of the size of the pasture fiber on the apparent density, porosity, and water absorption of the composite samples was insignificant. While the composite samples’ thermal conductivity increased with the pasture fiber’s size, the magnitude of this change was not significant.

Among the three types of pasture fiber, the composite sample with alfalfa exhibited the highest apparent density and thermal conductivity and the lowest water absorption and porosity. Conversely, the composite sample with millet-like sprangletop displayed the highest water absorption and the lowest thermal conductivity and apparent density. Additionally, the composite sample with sorghum had the highest porosity, indicating its strong ability to create pores.

Although increasing the content of pasture fiber can generate more pores, this abundance adversely affects the composite samples’ thermal conductivity.

According to the analysis of variance (ANOVA), pasture content had the highest contribution ratio to the composites’ thermal and physical properties.

Based on this study, incorporating pasture fibers less than 2% into slag bricks not only reduces the waste of pasture resources in pastoral areas, but also promotes the development of sustainable building materials with favorable thermal properties.

## Figures and Tables

**Figure 1 materials-17-03704-f001:**
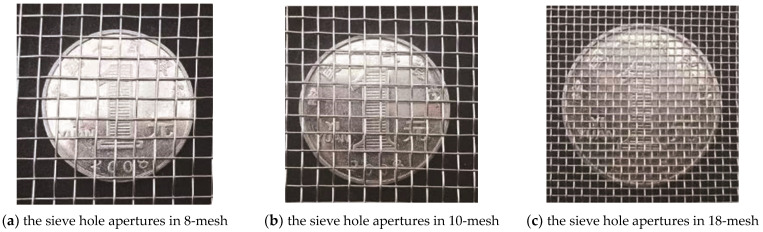
Three sieves with different mesh sizes of 8, 10, and 18.

**Figure 2 materials-17-03704-f002:**

Different sizes of alfalfa fiber: (**a**) 1 mm; (**b**) 2 mm; (**c**) 3 mm.

**Figure 3 materials-17-03704-f003:**
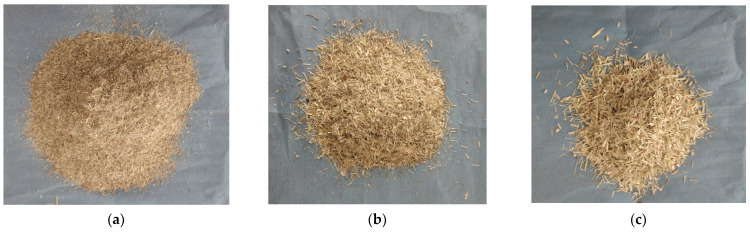
Different sizes of millet-like sprangletop: (**a**) 1 mm; (**b**) 2 mm; (**c**) 3 mm.

**Figure 4 materials-17-03704-f004:**
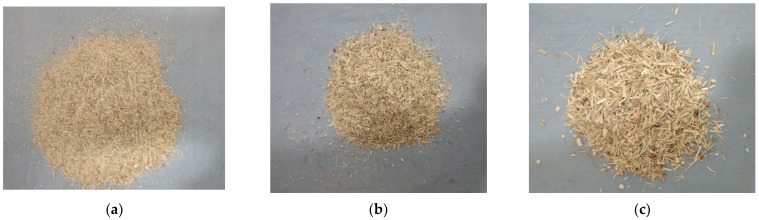
Different sizes of sorghum: (**a**) 1 mm; (**b**) 2 mm; (**c**) 3 mm.

**Figure 5 materials-17-03704-f005:**
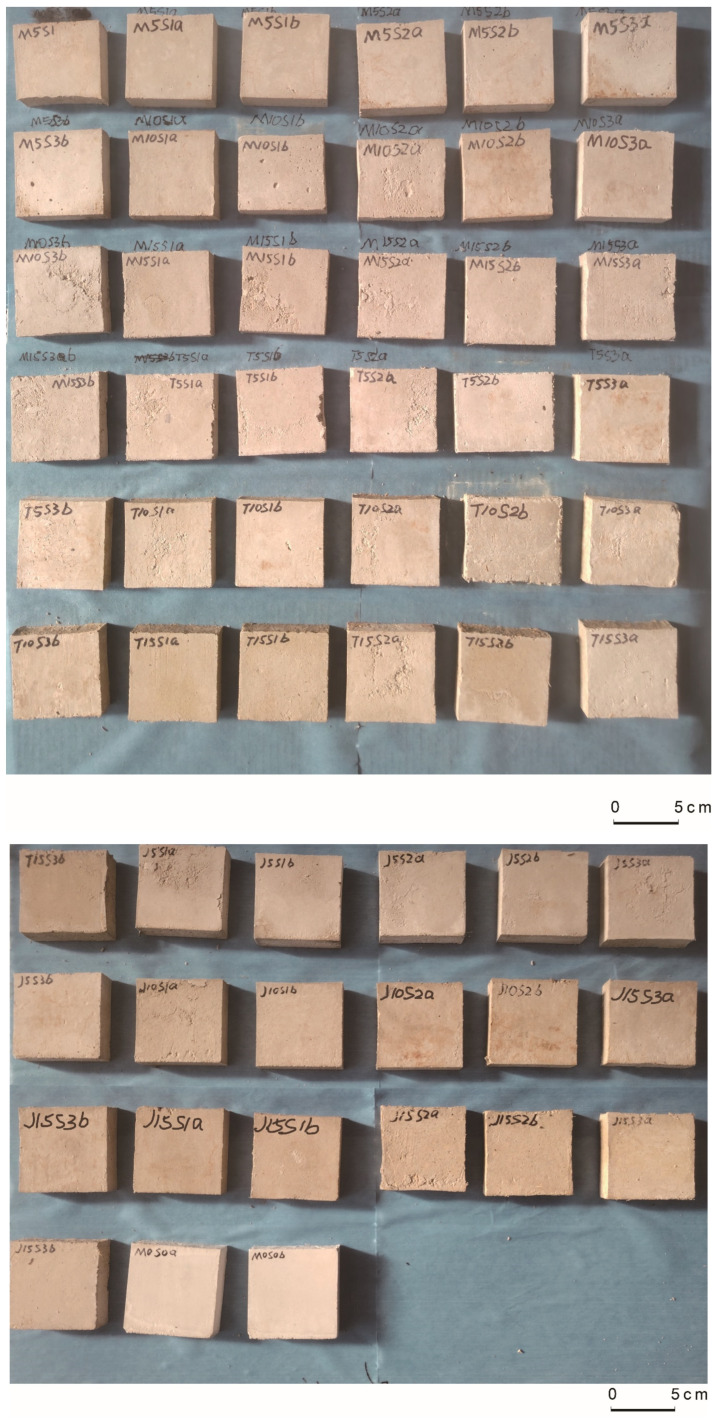
Images of the cured samples.

**Figure 6 materials-17-03704-f006:**
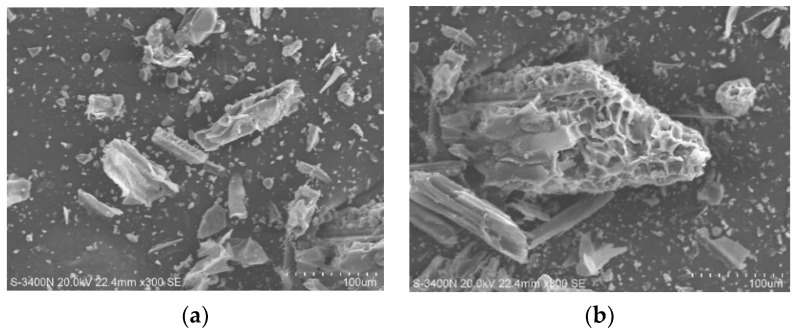
SEM image of millet-like sprangletop: (**a**) ×300; (**b**) ×500.

**Figure 7 materials-17-03704-f007:**
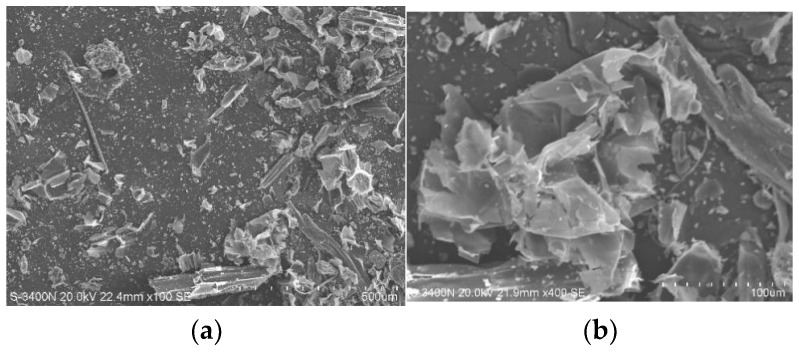
SEM image of sorghum: (**a**) ×300; (**b**) ×500.

**Figure 8 materials-17-03704-f008:**
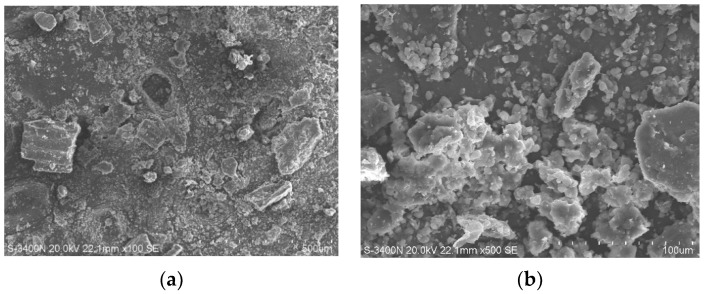
SEM image of alfalfa: (**a**) ×300; (**b**) ×500.

**Figure 9 materials-17-03704-f009:**
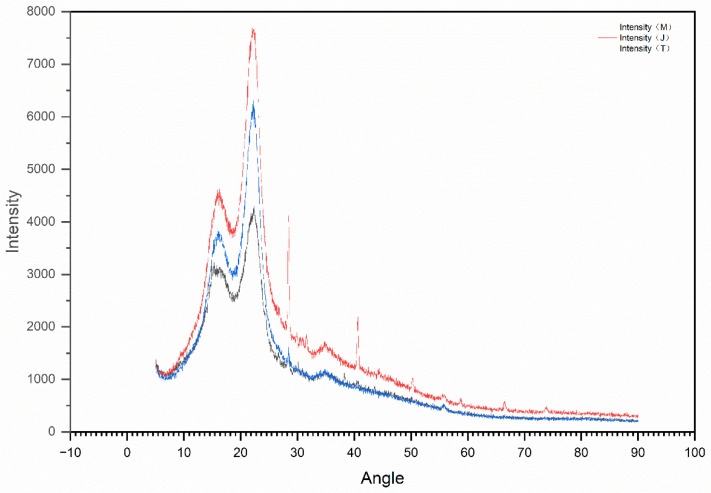
XRD of type of pasture fiber.

**Figure 10 materials-17-03704-f010:**
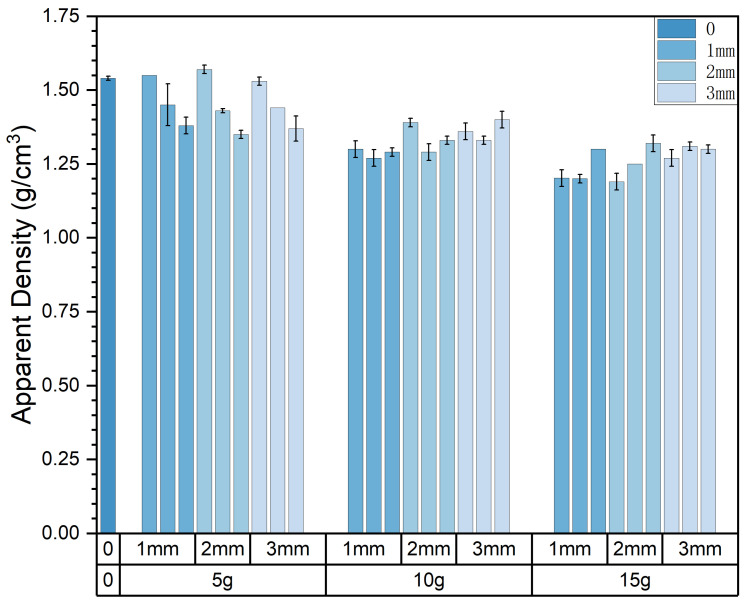
Content and size of pasture and apparent density of the composite samples. Color depth represents the added size of pasture, four colors from dark to light represent no pasture added, 1 mm, 2 mm and 3 mm.

**Figure 11 materials-17-03704-f011:**
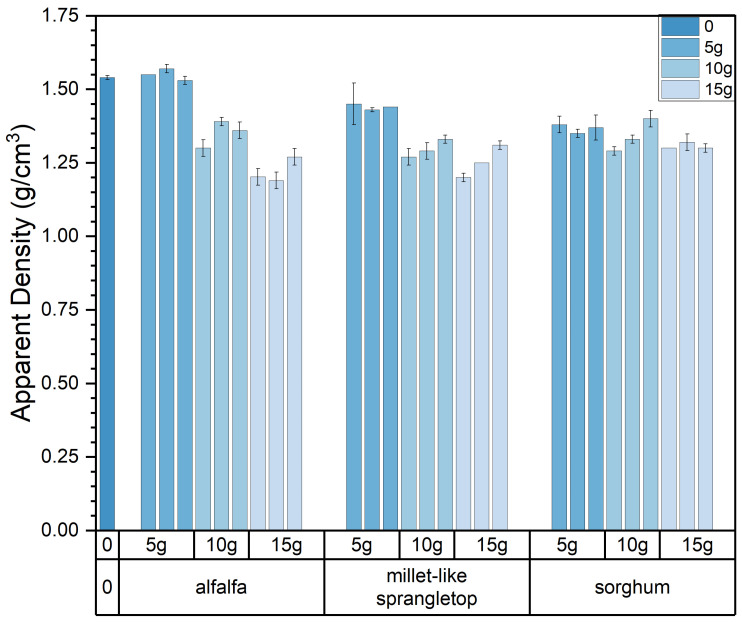
Type and content of pasture and apparent density of the composite samples. Color depth represents the added content of pasture, four colors from dark to light represent no pasture added, 5 g, 10 g and 15 g.

**Figure 12 materials-17-03704-f012:**
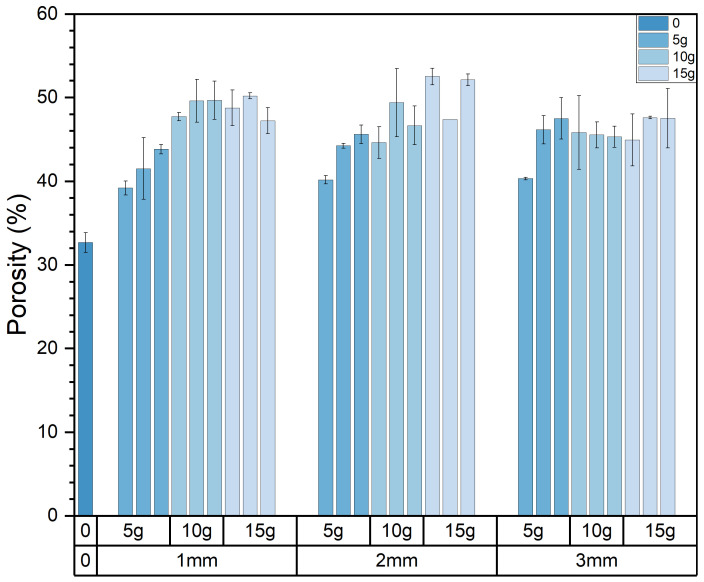
Size and content of pasture and porosity of the composite samples. Color depth represents the added content of pasture, four colors from dark to light represent no pasture added, 5 g, 10 g and 15 g.

**Figure 13 materials-17-03704-f013:**
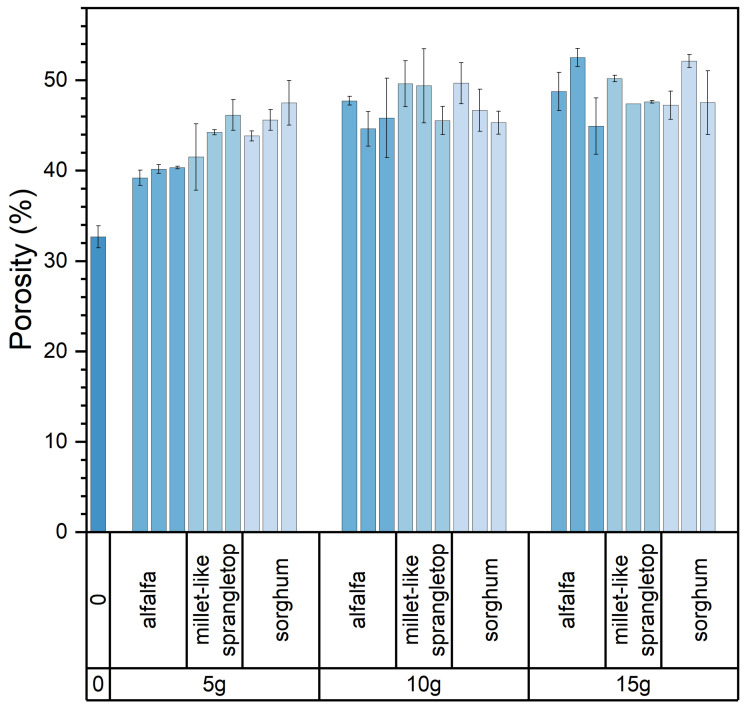
Content and type of pasture and porosity of the composite samples. Color depth represents the added type of pasture, four colors from dark to light represent no pasture added, alfalfa, millet-like sprangletop and sorghum.

**Figure 14 materials-17-03704-f014:**
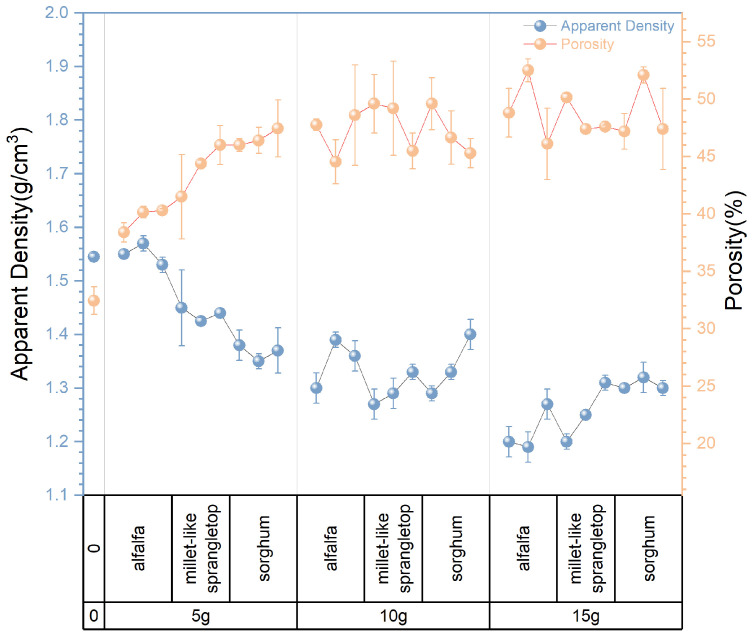
Porosity and apparent density.

**Figure 15 materials-17-03704-f015:**
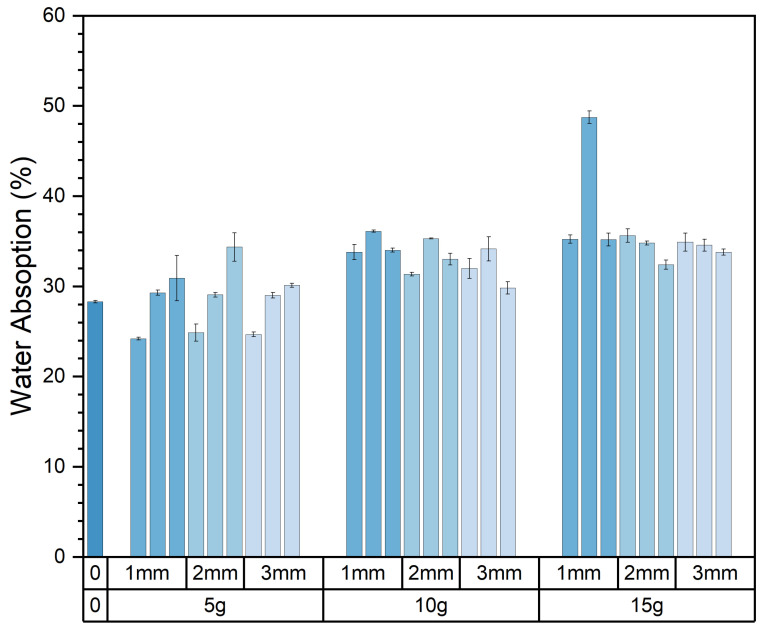
Content and size of pasture and water absorption of the composite samples. Color depth represents the added size of pasture, four colors from dark to light represent no pasture added, 1 mm, 2 mm and 3 mm.

**Figure 16 materials-17-03704-f016:**
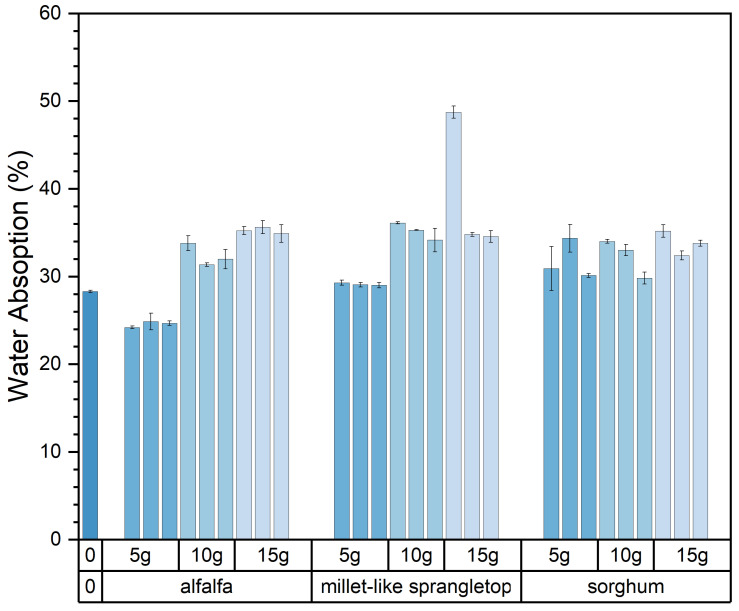
Type and content of pasture and water absorption of the composite samples. Color depth represents the added content of pasture, four colors from dark to light represent no pasture added, 5 g, 10 g and 15 g.

**Figure 17 materials-17-03704-f017:**
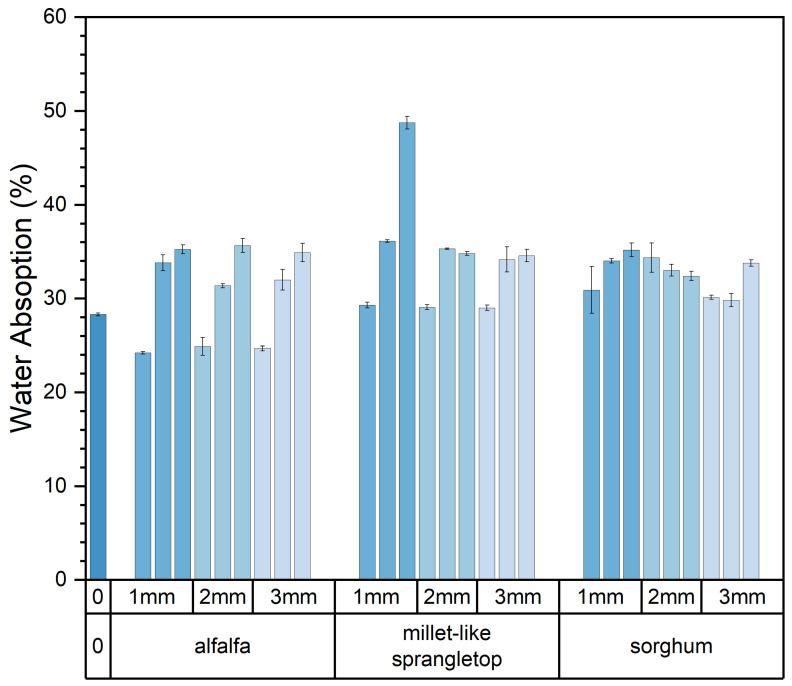
Type and size of pasture and water absorption of the composite samples. Color depth represents the added size of pasture, four colors from dark to light represent no pasture added, 1 mm, 2 mm and 3 mm.

**Figure 18 materials-17-03704-f018:**
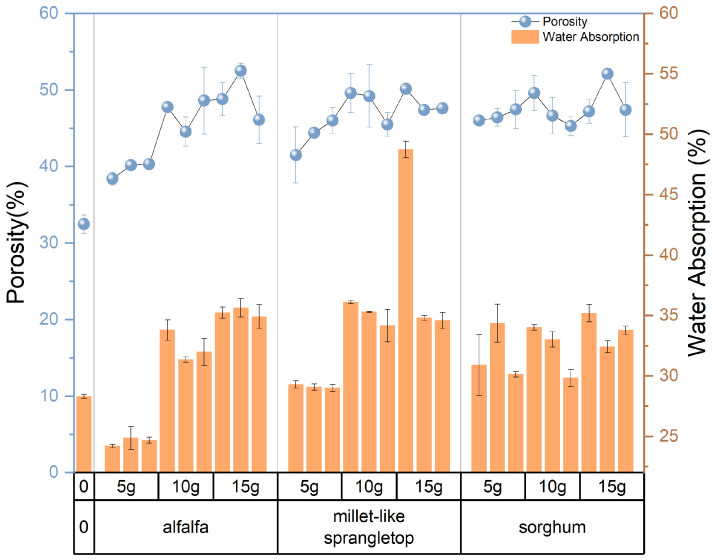
Porosity and water absorption.

**Figure 19 materials-17-03704-f019:**
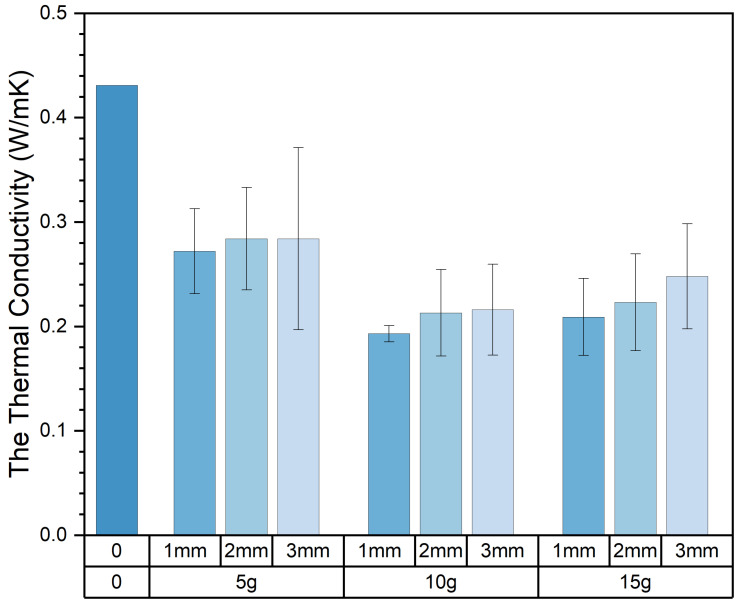
Content, size, and thermal conductivity. Color depth represents the added size of pasture, four colors from dark to light represent no pasture added, 1 mm, 2 mm and 3 mm.

**Figure 20 materials-17-03704-f020:**
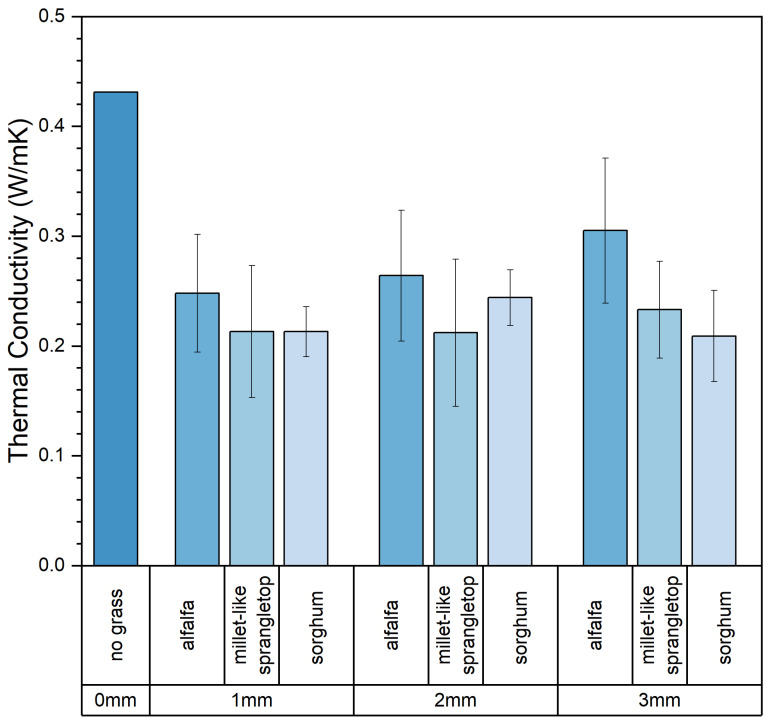
Size, type, and thermal conductivity. Color depth represents the added type of pasture, four colors from dark to light represent no pasture added, alfalfa, millet-like sprangletop and sorghum.

**Figure 21 materials-17-03704-f021:**
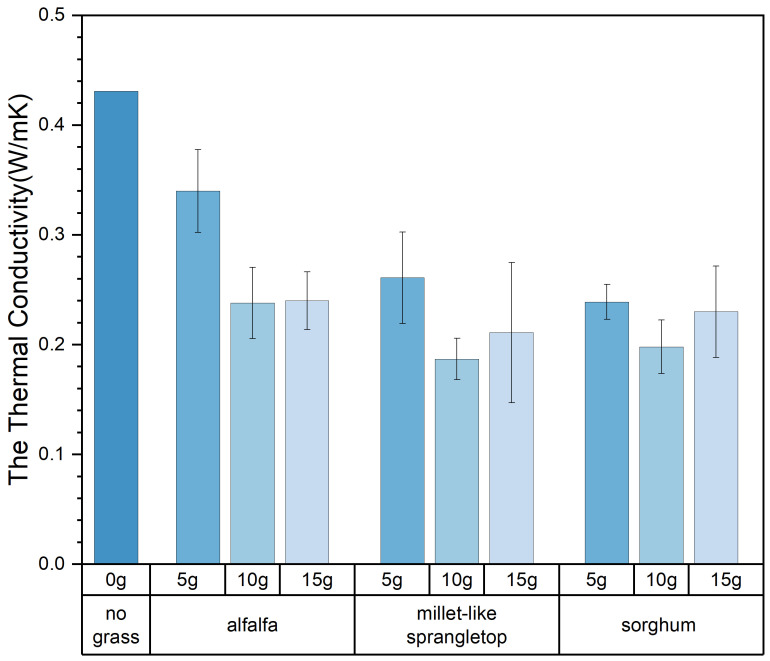
Type, content, and thermal conductivity. Color depth represents the added content of pasture, four colors from dark to light represent no pasture added, 5 g, 10 g and 15 g.

**Figure 22 materials-17-03704-f022:**
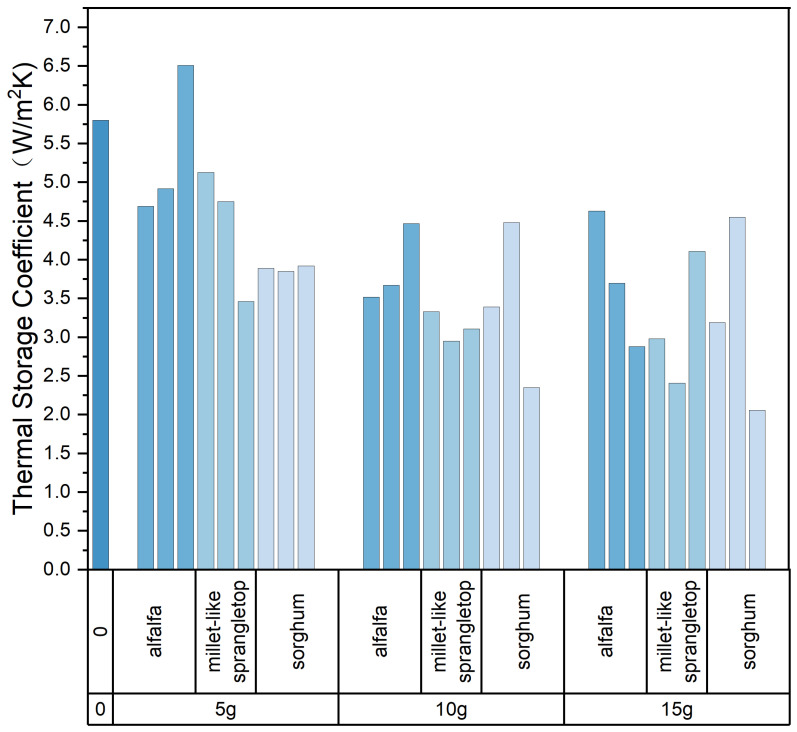
Thermal storage coefficient of composite samples. Color depth represents the added type of pasture, four colors from dark to light represent no pasture added, alfalfa, millet-like sprangletop and sorghum.

**Figure 23 materials-17-03704-f023:**
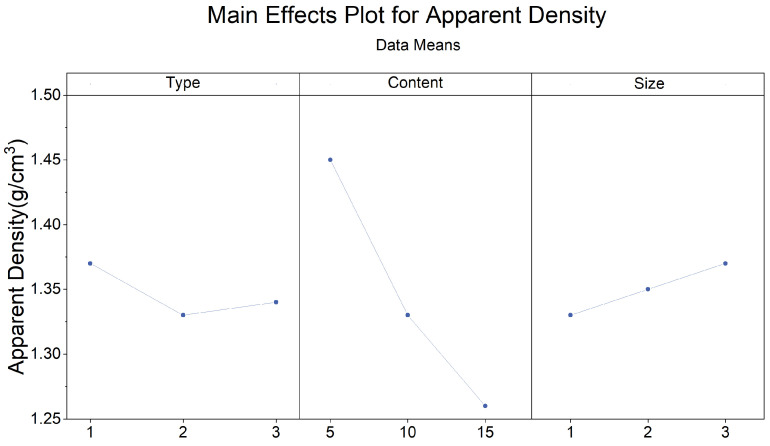
Main effects plot for apparent density. blue line represents the trend of the mean apparent density that varies with the type, content and size of pasture fibers. Numbers 1,2,3 of type mean different types of pasture fiber: alfalfa, millet-like sprangletop, and sorghum. This rule also applies to the subsequent images.

**Figure 24 materials-17-03704-f024:**
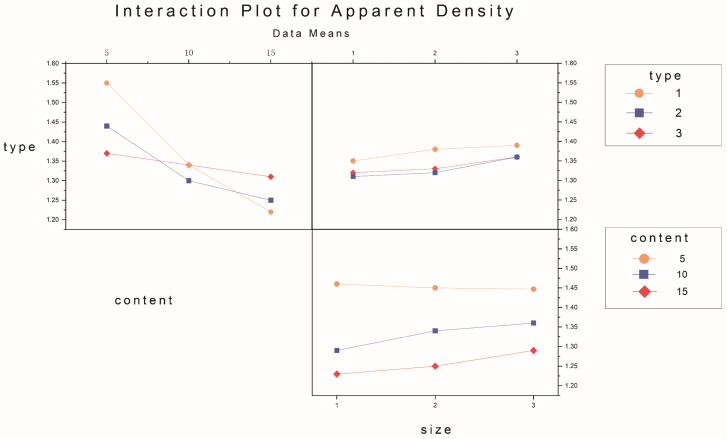
Interaction plot for apparent density.

**Figure 25 materials-17-03704-f025:**
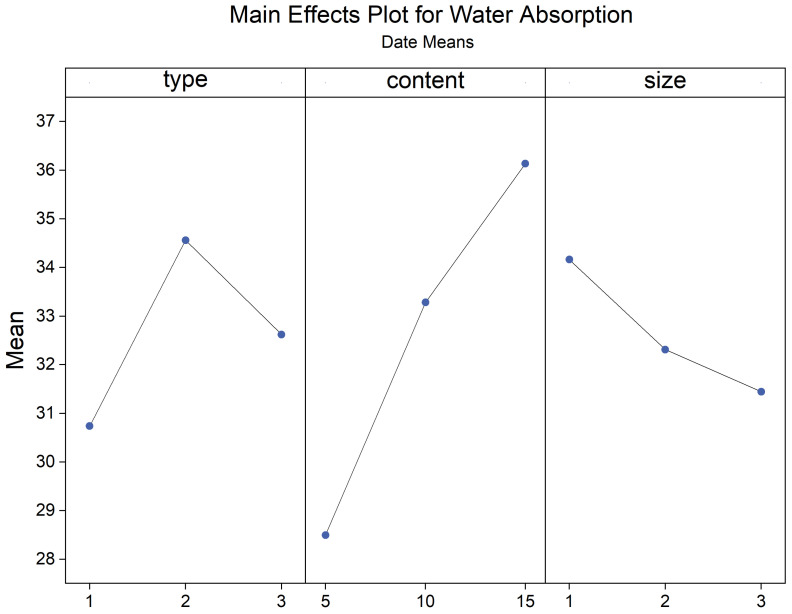
Main effects plot for water absorption. Blue line represents the trend of the mean water absorption that varies with the type, content and size of pasture fibers.

**Figure 26 materials-17-03704-f026:**
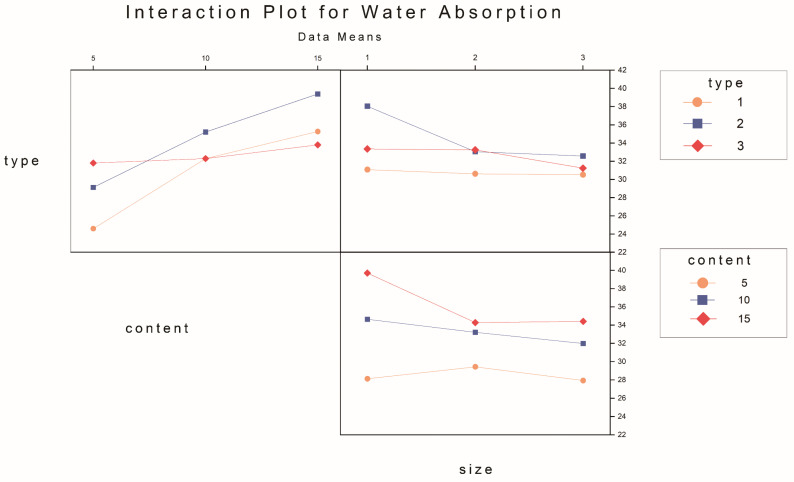
Interaction plot for water absorption.

**Figure 27 materials-17-03704-f027:**
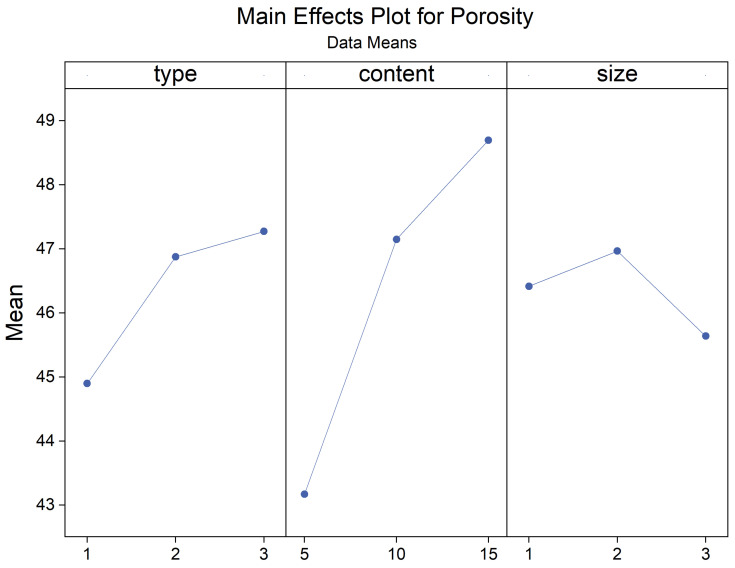
Main effects plot for porosity. Blue line represents the trend of the mean porosity that varies with the type, content and size of pasture fibers.

**Figure 28 materials-17-03704-f028:**
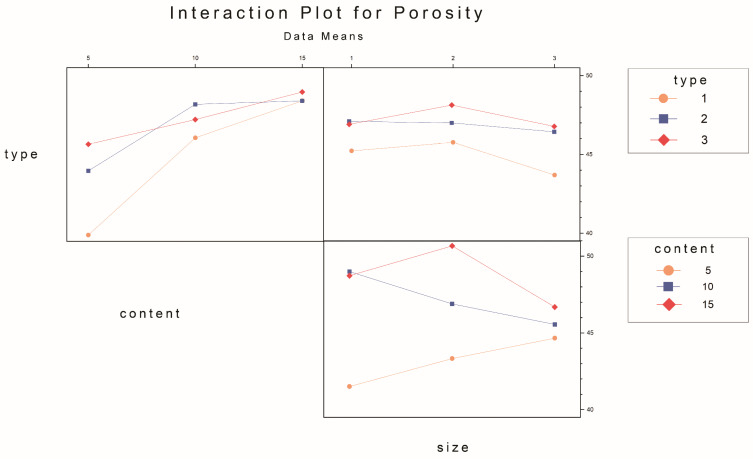
Interaction plot for porosity.

**Figure 29 materials-17-03704-f029:**
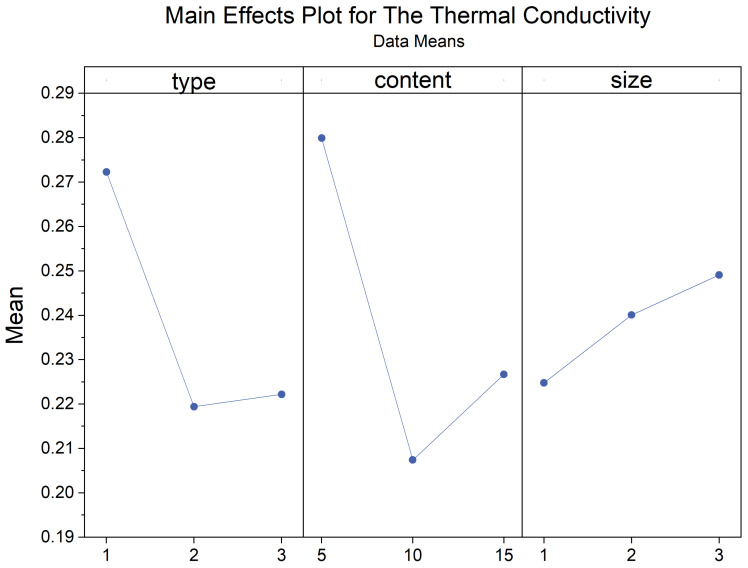
Main effects plot for thermal conductivity. Blue line represents the trend of the mean thermal conductivity that varies with the type, content and size of pasture fibers.

**Figure 30 materials-17-03704-f030:**
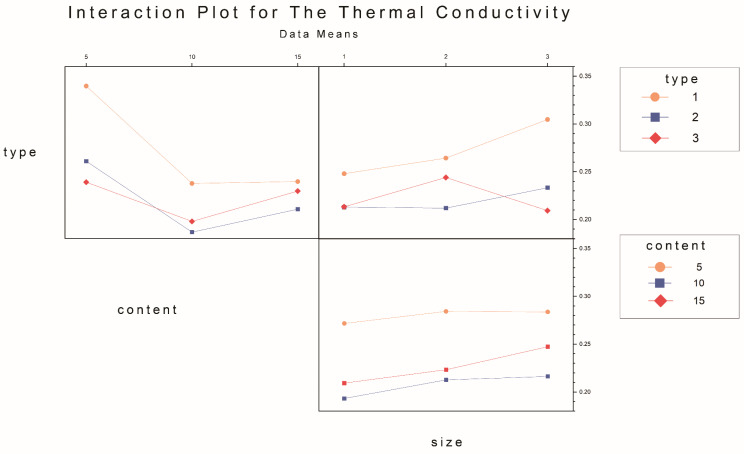
Interaction plot for thermal conductivity.

**Figure 31 materials-17-03704-f031:**
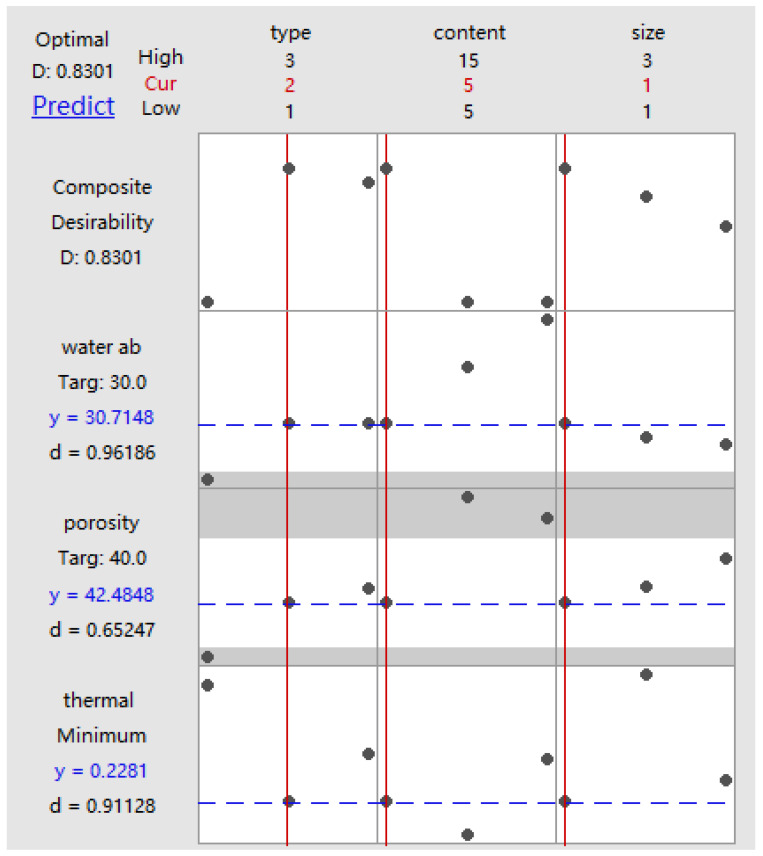
Multi-response prediction and desirability analysis of factors and response. Blue dashed line means predict value of water absorption, porosity and thermal minimum of samples; red line means the best value. Gray dot: The three points for each cell in this column represent the three levels of the categorical variable.

**Figure 32 materials-17-03704-f032:**
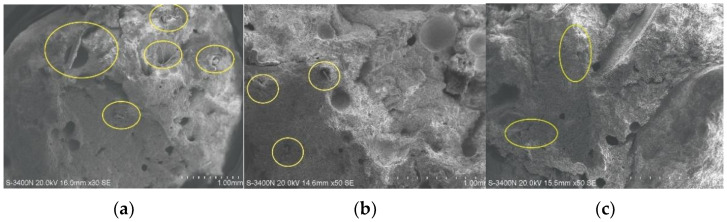
Internal voids in composite samples. The parts of yellow circles in three pictures represent the mixture of pastures and slag. (**a**) ×300, (**b**) ×500 and (**c**) ×1000.

**Figure 33 materials-17-03704-f033:**
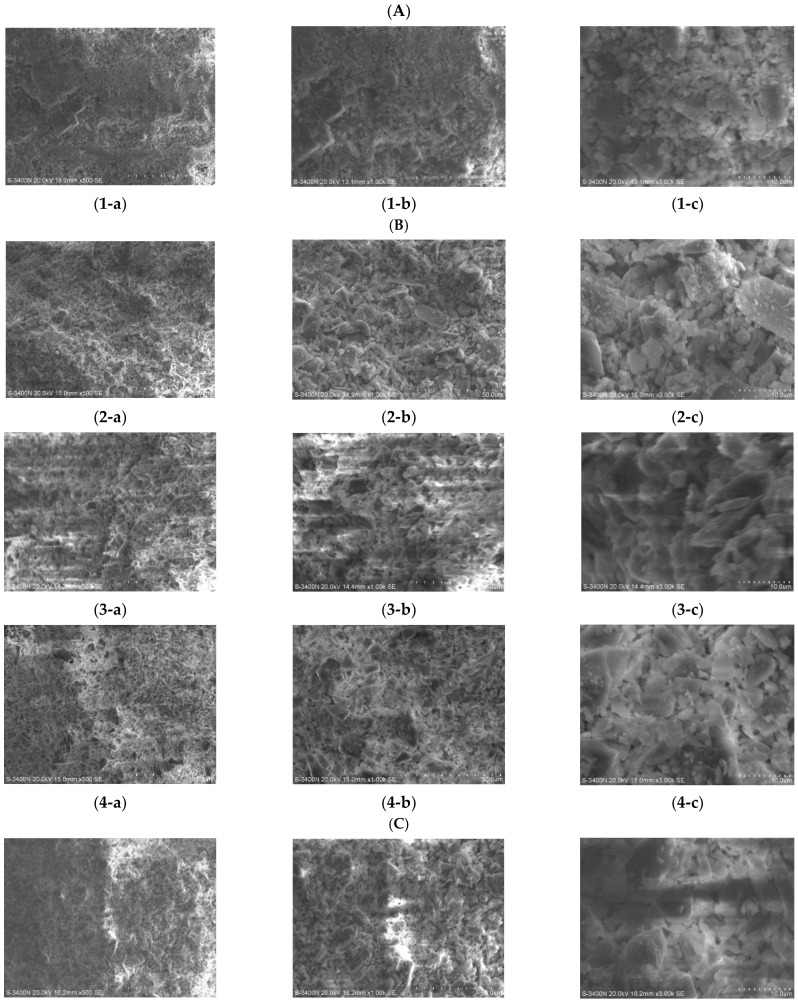
Microstructural images of composite samples with added pasture fiber slag. (**A**) Image of composite samples without pasture fiber addition: (**1-a**) ×500; (**1-b**) ×1000; (**1-c**) ×3000. (**B**) SEM images of composite samples with different contents of pasture fiber added. The composite samples with 5 g addition: (**2-a**) ×500; (**2-b**) ×1000; (**2-c**) ×3000. The composite samples with 10 g addition: (**3-a**) ×500; (**3-b**) ×1000; (**3-c**) ×3000. The composite samples with 15 g addition: (**4-a**) ×500; (**4-b**) ×1000; (**4-c**) ×3000. (**C**) The microscopic structure of composite samples with different types of pasture fiber added. (**5-a**–**5-c**) The microscopic structure of composite samples with millet-like sprangletop added: (**5-a**) ×500; (**5-b**) ×1000; (**5-c**) ×3000. (**6-a**–**6-c**) The microscopic structure of composite samples with sorghum added: (**6-a**) ×500; (**6-b**) ×1000; (**6-c**) ×3000. (**7-a**–**7-c**) The microscopic structure of composite samples with alfalfa added: (**7-a**) ×500; (**7-b**) ×1000; (**7-c**) ×3000.

**Table 1 materials-17-03704-t001:** The chemical composition of slag powder S105.

Chemical Compositions	CaO (g)	SiO_2_ (g)	Al_2_O_3_ (g)	SO_3_ (g)	Fe_2_O_3_ (g)	MgO (g)	Loss on Ignition (%)
	35.30	34.50	16.70	1.24	1.50	5.01	0.96

**Table 2 materials-17-03704-t002:** The relevant parameters of slag powder S105.

Specific Surface Area (m^2^/kg)	Flowability Ratio (%)	Activity Index 7d (%)	Activity Index 28d (%)	Density (g/cm^3^)	Moisture Content (%)
628.00	102.00	98.00	115.00	2.93	0.2

**Table 3 materials-17-03704-t003:** Full factorial experimental design.

Factor	Coded Value	Levels	Values
Type	M; J; T	3	1 (alfalfa), 2 (millet-like sprangletop), 3 (sorghum)
Size	S	3	1 (1 mm), 2 (2 mm), 3 (3 mm)
Content	CO	3	5, 10, 15

**Table 4 materials-17-03704-t004:** Experiment with mixed-proportion details.

Number	Type	Content (g)	Size (mm)	Alkali-Activated Granulated Blast Furnace Slag (g)	Sand (g)	Lime (g)	Water (g)
M5S1	Alfalfa	5	1 mm	165.22	55	20.62	130.66
M5S2	Alfalfa	5	2 mm	165.06	55.12	20.64	130
M5S3	Alfalfa	5	3 mm	165.09	55.01	20.68	130.62
M10S1	Alfalfa	10	1 mm	165.13	55.32	20.62	130.14
M10S2	Alfalfa	10	2 mm	165.16	55.13	20.79	130.72
M10S3	Alfalfa	10	3 mm	165	55.84	20.86	130.06
M15S1	Alfalfa	15	1 mm	165.02	55.09	20.27	131.67
M15S2	Alfalfa	15	2 mm	165.91	55.65	20.77	130.45
M15S3	Alfalfa	15	3 mm	165.7	55	20.99	130.8
J5S1	Millet-like sprangletop	5	1 mm	165.35	55.31	20.67	130
J5S2	Millet-like sprangletop	5	2 mm	165.01	55.31	20.69	130.55
J5S3	Millet-like sprangletop	5	3 mm	165.5	55.02	20.65	130.12
J10S1	Millet-like sprangletop	10	1 mm	165.4	55.3	20.93	130.01
J10S2	Millet-like sprangletop	10	2 mm	165.69	55.16	20.82	131.04
J10S3	Millet-like sprangletop	10	3 mm	165.77	55.59	20.54	130.91
J15S1	Millet-like sprangletop	15	1 mm	165.68	55.48	20.63	130.63
J15S2	Millet-like sprangletop	15	2 mm	165.53	55.71	20.69	130.63
J15S3	Millet-like sprangletop	15	3 mm	165.4	55.65	20.58	129.93
T5S1	Sorghum	5	1 mm	165.06	55.01	20.57	131.48
T5S2	Sorghum	5	2 mm	165.77	55.1	20.62	130.01
T5S3	Sorghum	5	3 mm	165.32	55	20.62	130.05
T10S1	Sorghum	10	1 mm	165.03	55.01	20.62	130.09
T10S2	Sorghum	10	2 mm	165.19	55.04	20.98	130.07
T10S3	Sorghum	10	3 mm	165	55.01	20.63	130.05
T15S1	Sorghum	15	1 mm	165.22	55.03	20.65	130.79
T15S2	Sorghum	15	2 mm	165.45	55.06	20.7	130.78
T15S3	Sorghum	15	3 mm	165.73	55.29	21.19	130.72
M0S0	No pasture	0	0 mm	165	55.02	20.62	130

**Table 5 materials-17-03704-t005:** Apparent density and true density.

Number	True Density (g/cm^3^)	Apparent Mass (g)	Apparent Volume (cm^3^)	Apparent Density (g/cm^3^)
M0S0	2.289	148.98	96.6	1.54
M5S1	2.515	142.86	92.475	1.55
M5S2	2.624	139.12	88.827	1.57
M5S3	2.564	143.33	93.8	1.53
M10S1	2.487	116.45	89.505	1.3
M10S2	2.510	118.35	85.291	1.39
M10S3	2.652	120.4	88.803	1.36
M15S1	2.346	109.53	91.12	1.202
M15S2	2.506	111.48	93.84	1.19
M15S3	2.306	107.31	84.286	1.27
J5S1	2.479	129.24	88.842	1.45
J5S2	2.564	130.18	90.821	1.43
J5S3	2.669	130.62	90.831	1.44
J10S1	2.520	110.09	86.873	1.27
J10S2	2.549	116.57	90.163	1.29
J10S3	2.442	114.83	86.526	1.33
J15S1	2.409	101.45	84.286	1.2
J15S2	2.376	111.24	88.842	1.25
J15S3	2.50	122.96	93.84	1.31
T5S1	2.557	125.94	91.125	1.38
T5S2	2.519	116.05	85.918	1.35
T5S3	2.61	126.95	92.475	1.37
T10S1	2.563	110.87	86.214	1.29
T10S2	2.494	115.28	86.853	1.33
T10S3	2.560	116.05	83.363	1.4
T15S1	2.464	116.69	89.495	1.3
T15S2	2.757	119.4	90.163	1.32
T15S3	2.477	111.04	85.248	1.3

**Table 6 materials-17-03704-t006:** The thermal properties of the composite samples.

Numbered	Th. Conductivity (W/m·K)	Th. Diffusivity (mm^2^/s)	Vol. Specif. Heat (MJ/m^3^·K)
M0S0	0.431	0.4	1.0742
M5S1	0.307	0.31	0.9844
M5S2	0.331	0.33	1.0038
M5S3	0.381	0.25	1.5261
M10S1	0.202	0.24	0.8429
M10S2	0.246	0.33	0.7524
M10S3	0.265	0.26	1.0324
M15S1	0.235	0.19	1.2493
M15S2	0.216	0.25	0.8709
M15S3	0.268	0.28	0.4155
J5S1	0.281	0.22	1.2827
J5S2	0.289	0.27	1.0728
J5S3	0.213	0.28	0.7711
J10S1	0.191	0.24	0.7953
J10S2	0.166	0.23	0.7199
J10S3	0.203	0.31	0.6511
J15S1	0.167	0.23	0.7276
J15S2	0.181	0.41	0.4396
J15S3	0.284	0.35	0.8152
T5S1	0.227	0.25	0.9121
T5S2	0.233	0.27	0.8743
T5S3	0.257	0.31	0.8215
T10S1	0.187	0.22	0.8453
T10S2	0.226	0.19	1.2193
T10S3	0.181	0.43	0.42
T15S1	0.226	0.37	0.6162
T15S2	0.273	0.26	1.0415
T15S3	0.19	0.62	0.3064

**Table 7 materials-17-03704-t007:** Mean of thermal conductivity, apparent density, and porosity.

Ratio of Pasture (%)	0	2%	4%	6%
Mean Thermal Conductivity (W/mK)	0.431	0.280	0.207	0.227
Mean Apparent Density (g/cm^3^)	1.54	1.45	1.33	1.26
Mean Porosity (%)	32.67	43.13	47.21	48.70

**Table 8 materials-17-03704-t008:** Factor analysis based on the complete factorial design.

Factors	Properties	Degrees of Freedom	SS	MS	F-Value	*p*-Value	Significance	%Contribution
Types	Thermal Conductivity	2	0.026	0.013	13.31	0.000	significant	18.17
	Thermal Diffusivity	2	0.029	0.014	9.25	0.001	significant	6.9
	Vol. Specific Heat	2	0.442	0.221	6.42	0.004	significant	11.33
	Apparent Density	2	0.019	0.100	21.66	0.000	significant	3.72
	Porosity	2	57.836	28.918	15.62	0.000	significant	9.69
	Water Absorption	2	131.49	65.747	19.40	0.000	significant	11.42
Size	Thermal Conductivity	2	0.004	0.002	1.91	0.164	insignificant	2.6
	Thermal Diffusivity	2	0.078	0.039	25.19	0.000	significant	18.77
	Vol. Specific Heat	2	0.207	0.103	3.00	0.063	insignificant	5.30
	Apparent Density	2	0.015	0.008	16.78	0.000	significant	2.88
	Porosity	2	16.073	8.036	4.34	0.021	significant	2.69
	Water Absorption	2	69.35	34.675	10.23	0.000	significant	6.02
Content	Thermal Conductivity	2	0.053	0.027	27.47	0.000	significant	37.48
	Thermal Diffusivity	2	0.036	0.018	11.60	0.000	significant	8.65
	Vol. Specific Heat	2	0.800	0.400	11.60	0.000	significant	20.47
	Apparent Density	2	0.341	0.170	379.95	0.000	significant	65.18
	Porosity	2	292.754	146.377	79.05	0.000	significant	49.07
	Water Absorption	2	536.18	268.091	79.12	0.000	significant	46.58
Type × content	Thermal Conductivity	4	0.018	0.004	4.54	0.005	significant	12.4
	Thermal Diffusivity	4	0.071	0.018	11.57	0.000	significant	17.25
	Vol. Specific Heat	4	0.170	0.043	1.24	0.314	insignificant	4.36
	Apparent Density	4	0.115	0.029	64.12	0.000	significant	22.00
	Porosity	4	61.376	15.344	8.29	0.000	significant	10.29
	Water Absorption	4	161.26	40.314	11.9	0.000	significant	14.01
Type × size	Thermal Conductivity	4	0.007	0.002	1.93	0.127	insignificant	5.27
	Thermal Diffusivity	4	0.112	0.028	18.13	0.000	significant	27.02
	Vol. Specific Heat	4	0.893	0.223	6.48	0.001	significant	22.88
	Apparent Density	4	0.002	0.001	1.19	0.331	insignificant	0.41
	Porosity	4	6.005	1.501	0.81	0.527	insignificant	1.01
	Water Absorption	4	58.99	14.747	4.35	0.006	significant	5.12
Content × size	Thermal Conductivity	4	0.000	0.000	0.07	0.990	insignificant	2.0
	Thermal Diffusivity	4	0.035	0.009	5.62	0.001	significant	8.37
	Vol. Specific Heat	4	0.186	0.047	1.35	0.271	insignificant	4.77
	Apparent Density	4	0.015	0.0037	8.19	0.000	significant	2.81
	Porosity	4	97.708	24.427	13.19	0.000	significant	16.38
	Water Absorption	4	75.19	18.797	5.55	0.001	significant	6.53

**Table 9 materials-17-03704-t009:** Parameter settings for multi-response optimization prediction.

Response	Goal	Lower	Target	Upper	Weight	Importance
Water Absorption (%)	Target	24.2	30	48.74	1	1
Thermal Conductivity (W/mK)	Minimum	0.166	0.166	0.381	1	1
Porosity (%)	Target	39.2	40	47.15	1	1

**Table 10 materials-17-03704-t010:** Optimizing blend proportions and experimental results.

Particulars	Water Absorption (%)	Porosity (%)	Thermal Conductivity (W/mK)
Response Prediction (millet-like sprangletop 5 g, 1 mm)	30.71	42.49	0.23
Range	26.37–35.06	49.27–45.70	0.14–0.32
Experimental Results (millet-like sprangletop 5 g, 1 mm)	29.29	41.51	0.281

## Data Availability

The authors will provide data upon request.

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
