# Peer review of "Study on the Effects of Pasture Fiber on Thermal Properties of Slag Bricks"

_materials, 2024, doi:10.3390/ma17153704_

Round 1

Reviewer 1 Report

Comments and Suggestions for Authors

This study examines the impact of various pasture fibers on the thermal properties of slag bricks, utilizing slag as a binder, sand as an aggregate, and pasture fibers as additives. Using Minitab 18 software, the analysis reveals that increasing pasture fiber content initially decreases, then improves thermal performance. Lower thermal conductivity corresponds with lower density and higher porosity. The findings suggest that incorporating pasture fibers into slag bricks reduces waste and enhances sustainable building materials. While the use of natural fibers is interesting, the durability remains questionable. Durability tests are lacking, specifically regarding how moisture and temperature fluctuations would affect such a composite. Some general questions:

  1. In the introduction, justify the originality of your work by comparing it with similar studies.
  2. In section 2.1, Raw Materials, ensure chemical formulas are correct (numbers should be lowercase).
  3. Are these your results (Table 1)? Please add the measuring technique (e.g., XRF?).
  4. Correct the numbering of equations.
  5. Pictures 9 and 10 do not fit in a scientific article.
  6. All figures above 11 are very poorly visible; add the standard error of the measurements.
  7. In the Factor Analysis and Optimization of Composite Sample Properties section, draw your own figures to improve visibility.
Comments on the Quality of English Language

Moderate editing of English language required

Author Response

Comments 1: In the introduction, justify the originality of your work by comparing it with similar studies.

Response 1: Thank you for pointing this out. We agree with this comment. Therefore, About Introduction, I consulted a large amount of relevant literature in the second and third phases, I found these studies focus on incorporating natural plant fibers into clay bricks and concrete, and researches on adding pasture fiber are not easily found. The few articles available are focus on the mechanical properties of pasture composite materials.  Therefore, from 72—78, I indicated that there is a lack of research on using pasture as an additive in construction materials. All of them focus on the mechanical properties of composite material. 

Comments 2: In section 2.1, Raw Materials, ensure chemical formulas are correct (numbers should be lowercase).

Response 2: Agree. We have revised them to emphasize this point. I have already modified them at 111,112,113, and 115.

Comments 3: Are these your results (Table 1)? Please add the measuring technique (e.g., XRF?).

Response 3: Agree. I have already modified them to 109,110.

Comments 4: Correct the numbering of equations.

Response 4: Agree. I have already modified them at 240,254,263, and 471.

Comments 5: Pictures 9 and 10 do not fit in a scientific article.

Response 5: Agree. I have already cut them.

Comments 6: All figures above 11 are very poorly visible; add the standard error of the measurements.

Response 6: Agree. I have already redesigned these graphics and added the standard error of the measurement.

Comments 7: In the Factor Analysis and Optimization of Composite Sample Properties section, draw your own figures to improve visibility.

Response 7: Agree. I have already redesigned these graphics including the section on Factor analysis and Optimization of Composite Properties.

4. Response to Comments on the Quality of English Language

Point 1: Moderate editing of the English language required

Response 1: I have corrected language issues, but due to not being a native speaker, I may not be able to identify all problems. If there are still issues, I may seek help from a language institution.

Reviewer 2 Report

Comments and Suggestions for Authors

The article has the potential to be quite interesting, but it still requires thorough work. Below are my initial comments:

- lines 107-111 - the authors do not use upper and lower subscripts - it is necessary to check the entire article in this respect

- lines 96, 98, 116, etc. - the authors miswrote the units - no spaces between the numerical value and the unit - it is necessary to check the entire article in this respect

- lines 109-110 - an incorrect record of the chemical formula of the presented phases - it is necessary to check the entire article in this respect

With such diversity in the shape of the feed, the sieving process becomes a crucial aspect of the research. I'm curious about the shape of the meshes you used in the sieving process. Could you please provide more information about the sieving process, the sieves used, vibration frequencies, sieving time, etc.? It would be great to see the device and sieves in the photo. Did you notice any differences in this respect depending on the size of the material being screened? How did you determine the screening time?

Figures 1-3 do not have a scale - not even a ruler, which is crucial for determining the differences in size between the presented elements. Please complete this detail.

In Figure 4, some samples are signed, and some are not. Why? The photos also lack scale.

I noticed that the caption for Figure 4 is not complete. It currently reads 'Figure 4 cured sample '. A complete and descriptive caption is important as it provides crucial context for the reader's understanding. Could you please review and complete it?

At this stage, I am not checking the publications further. The authors should re-read the entire article they wrote, fill in the gaps, and correct any irregularities—even those required by the MDPI Publishing House. In this form, the article is unsuitable for publication. 

Comments on the Quality of English Language

No comment on this matter. No objections.

Author Response

Comments 1: lines 107-111 - the authors do not use upper and lower subscripts - it is necessary to check the entire article in this respect

Response 1: Thank you for pointing this out. We agree with this comment. I have reviewed and corrected any issues regarding format, units, and chemical formulas in the text.

Comments 2: lines 96, 98, 116, etc. - the authors miswrote the units - no spaces between the numerical value and the unit - it is necessary to check the entire article in this respect

Response 2: Agree. We have revised them to emphasize this point. 1.     I have reviewed and corrected any issues regarding format, units, and chemical formulas in the text.

Comments 3: - lines 109-110 - an incorrect record of the chemical formula of the presented phases - it is necessary to check the entire article in this respect

Response 3: Agree. I have already modified them to 109,110.

Comments 4: With such diversity in the shape of the feed, the sieving process becomes a crucial aspect of the research. I'm curious about the shape of the meshes you used in the sieving process. Could you please provide more information about the sieving process, the sieves used, vibration frequencies, sieving time, etc.? It would be great to see the device and sieves in the photo. Did you notice any differences in this respect depending on the size of the material being screened? How did you determine the screening time?

Response 4: Agree. I have already added them. The following is my interpretation of how to separate the size of pasture fibers:

These sieves used for the experiment were customized experimental sieves from JiuFeng Sieve, The Figure shows the sieve hole apertures in 8 mesh, 10 mesh, and 18 mesh shapes. To separate the pasture sizes, the collected pasture was poured into an 8-mesh sieve and shaken manually from side to side for approximately 30 seconds to sift the pasture. Then, the sifted pasture was transferred to a 10-mesh sieve, which was shaken horizontally by the same person for about 3 minutes. After three minutes, the pasture on top of the sieve was set at the experimental size of 3 mm. The same procedure was repeated, and after three minutes, the pasture on top of the sieve was of the experimental size of 2 mm and the sieved pasture was of the experimental size of 1 mm.120-133.

Comments 5: Figures 1-3 do not have a scale - not even a ruler, which is crucial for determining the differences in size between the presented elements. Please complete this detail.

Response 5: Agree. For Figure 1-3, I did not realize the importance of having a scale in the images. Unfortunately, as the raw materials have been used up, I am unable to provide pictures with a scale. However, I have provided a detailed description of how to separate pasture fibers of different sizes, and I hope this will be helpful.

Comments 6: In Figure 4, some samples are signed, and some are not. Why? The photos also lack scale.

Response 6: I have already modified the picture. During the curing phase, these samples need to be watered regularly to ensure sufficient humidity. To maintain uniform humidity on each surface, the samples are flipped over. Some surfaces may not display the labels due to this flipping process; in reality, they are on the opposite side. I have now labeled them accordingly.

I have already redesigned these graphics and added the standard eoor of the measurement.

Comments 7: I noticed that the caption for Figure 4 is not complete. It currently reads 'Figure 4 cured sample '. A complete and descriptive caption is important as it provides crucial context for the reader's understanding. Could you please review and complete it?

Response 7: Agree. The caption for Figure 4 has changed, it is “The image of the cured samples”. 172.

4. Response to Comments on the Quality of English Language

Point 1: No comment on this matter. No objections.

Response 1: Nothing.

Round 2

Reviewer 1 Report

Comments and Suggestions for Authors

The manuscript was improved

Comments on the Quality of English Language

It is okay

Reviewer 2 Report

Comments and Suggestions for Authors

I have no more comments or suggestions.